# Volume-Aware Distance for Robust Similarity Learning

**Shuo Chen** [1]  **Chen Gong** [2]  **Jun Li** [2]  **Jian Yang** [2]

## Abstract

Measuring the similarity between *data points* plays a vital role in lots of popular representation learning tasks such as *metric learning* and *contrastive learning*. Most existing approaches utilize *point-level* distances to learn the *point-to-point* similarity between pairwise instances. However, since the finite number of training data points cannot fully cover the whole *sample space* consisting of an infinite number of points, the *generalizability* of the learned distance is usually limited by the *sample size*. In this paper, we thus extend the conventional form of data point to the new form of *data ball* with a predictable volume, so that we can naturally generalize the existing point-level distance to a new *volume-aware distance* (VAD) which measures the *field-to-field* geometric similarity. The learned VAD not only takes into account the relationship between observed instances but also uncovers the similarity among those *unsampled* neighbors surrounding the training data. This practice significantly enriches the coverage of sample space and thus improves the model generalizability. Theoretically, we prove that VAD tightens the *error bound* of traditional similarity learning and preserves crucial *topological properties*. Experiments on multi-domain data demonstrate the superiority of VAD over existing approaches in both *supervised* and *unsupervised* tasks.

## 1. Introduction

*Similarity learning* has been a longstanding focus of research, aiming to learn to measure pairwise similarity between instances. The learned *similarity metric* and *feature representation* can serve as essential components for down-

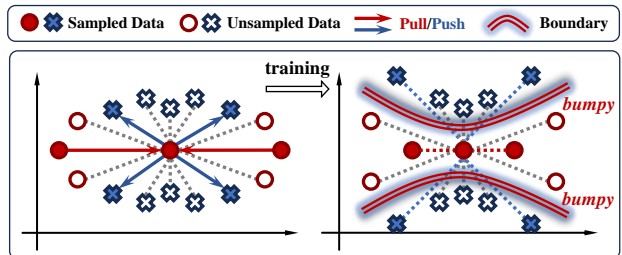

**(a). Concept illustration of the traditional point-level distance**

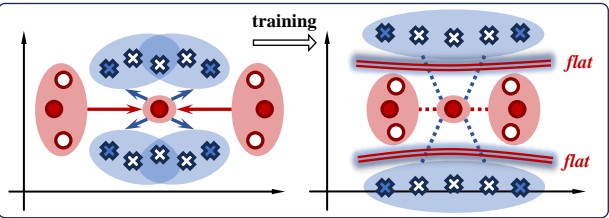

**(b). Concept illustration of our *volume-aware distance* (VAD)**

*Figure 1.* A quick comparison between the traditional point-level distance and our proposed VAD. By considering the volume awareness, our VAD can obtain the flatter decision boundary for better generalizability than the traditional distance.

stream recognitions (Weinberger et al., 2006; Xing et al., 2002; Liu et al., 2018; Yan et al., 2024). Over the past decades, similarity learning has achieved remarkable success, showing its potency in diverse representative tasks, such as *classification* (Kaya & Bilge, 2019), *clustering* (Zhong et al., 2020), and *retrieval* (Kou et al., 2022).

Typically, similarity learning algorithms are provided with (pseudo) supervision in the form of pairwise relationships (e.g., similar or dissimilar) derived from the training data (Sohn, 2016; Feng et al., 2023). Then the learning algorithms pull similar instances closer while pushing dissimilar ones apart, and the overarching goal of similarity learning is to make the learned similarity metric robust and generalizable when applied to the complex test phase with unseen data (Yan et al., 2023b; Kaya & Bilge, 2019).

Substantial progress has been made toward this goal through the design of various loss functions (Liu & Tsang, 2015; Sohn, 2016; Oh Song et al., 2016; Chen et al., 2020) and plentiful regularization terms (Chen et al., 2021; Yan et al., 2022b; Wang & Qi, 2023; Chen et al., 2024a). These methods refine the sampling process and constrain the hypothesis

[1]School of Intelligence Science and Technology, Nanjing University, China [2]School of Computer Science and Technology, Nanjing University of Science and Technology, China. Correspondence to: Shuo Chen <shuo.chen@nju.edu.cn>.

*Proceedings of the $42^{st}$ International Conference on Machine Learning*, Vancouver, Canada. PMLR 267, 2025. Copyright 2025 by the author(s).

space to improve learning performance. However, the *generalizability* of similarity learning remains limited by the *sample size*, as *finite* sampled *data points* (i.e., instances) are unable to sufficiently cover the whole sample space which contains *infinite* unsampled points. The coverage insufficiency is particularly hard in similarity learning due to considering the *Cartesian product* (i.e., the pairwise relationship) of the sample space (Feng et al., 2021; Wu et al., 2022a). As a result, the *decision boundary* of similarity metric learned from the scattered data points may lack the *flatness* (see Fig. 1(a)) needed for generalizable classification. This practically leads to unsatisfactory performance in some noisy and difficult recognition scenarios for both supervised tasks (e.g., *metric learning*) and unsupervised/self-supervised tasks (e.g., *contrastive learning*).

Given that the entire sample space usually assumes a specific volume (e.g., the widely adopted hypercube/hypersphere with feature normalization (Xu et al., 2022a; Yan et al., 2021)), it is inherently challenging to adequately fill in this space by using only volumeless data points. To address this issue, we introduce a new *measure-head network* regularized by *volume expansion strategy* to predict the volume of each data point (i.e., extending to *data balls*), thereby effectively considering pairwise relationships within the unsampled neighborhoods surrounding training data points (see Fig. 1(b)). This allows us to define a novel *volume-aware distance* (VAD) to measure the *geometric proximity* between volume-predictable data balls, successfully learning a reliable similarity metric with better coverage of the sample space. Theoretically, we establish the geometric soundness of VAD and derive a tighter error bound. Extensive experiments conducted across multiple domains demonstrate the effectiveness and superiority of our approach. Our main contributions are summarized below: **1)** We propose a novel VAD metric for robust similarity learning, supported by comprehensive theoretical analyses that ensure its soundness and effectiveness; **2)** We build a new similarity learning framework incorporating a measure-head network that adaptively predicts instance volumes for reliable similarity determination; **3)** The experiments on both the supervised and unsupervised tasks successfully validate the superiority of our method over the state-of-the-art approaches.

## 2. Background & Related Work

We briefly review the research related to this paper.

**Notations.** We write matrices, vectors, and mappings as bold uppercase characters, bold lowercase characters, and calligraphy characters, respectively. We denote the training dataset $\mathscr{X} = \{\boldsymbol{x}_i \in \mathbb{R}^d | i = 1, 2, \ldots, N\}$ where $d$ is the data dimensionality and $N$ is the total number of instances.

### 2.1. Metric Learning & Contrastive Learning

In supervised scenarios, similarity learning is commonly referred to as metric learning (Xing et al., 2002; Yan et al., 2023a; Kaya & Bilge, 2019), where the pairwise relationships are provided through human supervision. Metric learning has been extensively studied in both linear models (Davis et al., 2007; Zadeh et al., 2016; Chen et al., 2019) and nonlinear deep neural network based models (Chu et al., 2020; Yan et al., 2023b; Furusawa, 2024). A lot of existing research has focused on designing novel loss functions to enrich sampling results, e.g., *triplet loss* (Ge, 2018), *N-pairs loss* (Sohn, 2016), *circle loss* (Sun et al., 2020b), etc.

In unsupervised scenarios, self-supervised contrastive learning has garnered significant attention due to its competitive performance compared with the fully supervised approach (Chen et al., 2020; Yan et al., 2022a). This approach adopts the similarity metric framework from traditional metric learning and also trains an encoder network in a pairwise manner. Contrastive learning builds the positive (similar) data pairs by pulling each instance closer to its *data augmentation* and creates negative (dissimilar) data pairs by pushing each instance away from others (Chuang et al., 2020; Yan et al., 2022a; Tian et al., 2020; Chen et al., 2021).

Both supervised and unsupervised methods rely on point-to-point similarity metrics. We propose a novel metric that considers field-to-field relationships within the sample space.

### 2.2. Regularization & Augmentation

The primary goal of regularization is to enhance the generalizability of learning algorithms. Early works minimize the $\ell_2/\ell_1$-norm (Arpit et al., 2016; Yang et al., 2011), or nuclear norm (Chen et al., 2022a; Dong et al., 2014) of learning parameters to constrain the hypothesis space within interpretable regions. Recent works such as *dropout* (Baldi & Sadowski, 2013), *batch normalization* (Bjorck et al., 2018), and *mixup* (Zhang et al., 2018) focus on regularizing the flat decision boundary to achieve smooth generalization.

As an implicit form of regularization, data augmentation is highly effective not only in similarity learning but also across a broad spectrum of representation learning (Steiner et al., 2021; You et al., 2020; Zheng et al., 2021). It leverages human prior knowledge to quickly expand the training dataset, enabling models to better capture neighborhood relationships among sampled data points. Notably, recent advancements, such as set-level similarity (Wang et al., 2022), automatic augmentation (You et al., 2021), and adversarial augmentation (Lim et al., 2023), have further enriched original data and extended sample space coverage.

In this paper, we consider a new metric that expands sample space coverage and regularizes the learning process without relying on additional augmented data.

# 3. Methodology

In this section, we formulate our proposed new VAD metric and the corresponding learning objective.

## 3.1. New Metric and New Framework

The projected Euclidean distance has been a prevalent similarity metric in existing research (Liu et al., 2018). For instances $\boldsymbol{x}, \widehat{\boldsymbol{x}} \in \mathbb{R}^d$, this metric is generally defined as:

$$d_{\boldsymbol{\varphi}}(\boldsymbol{x}, \widehat{\boldsymbol{x}}) = \|\boldsymbol{\varphi}(\boldsymbol{x}) - \boldsymbol{\varphi}(\widehat{\boldsymbol{x}})\|_2, \quad (1)$$

where $\boldsymbol{\varphi} : \mathbb{R}^d \to \mathbb{R}^m$ represents a feature embedding learned by an encoder network (e.g., *ResNet* (He et al., 2016) and *ViT* (Han et al., 2022)), and the corresponding embedding result is sometimes further normalized, namely, for any $\boldsymbol{x} \in \mathbb{R}^d$, $\boldsymbol{\varphi}(\boldsymbol{x}) = \widehat{\boldsymbol{\varphi}}(\boldsymbol{x})/\|\widehat{\boldsymbol{\varphi}}(\boldsymbol{x})\|_2$.

**Volume-Aware Distance.** As the above conventional metric neglects the important information from the neighborhood fields of $\boldsymbol{x}$ and $\widehat{\boldsymbol{x}}$ during its distance calculation, here we want to extend it to a volume-aware form. To be specific, we build a measure function $\mathcal{V} : \mathbb{R}^d \to \mathbb{R}^+$ to characterize the volume value $\mathcal{V}(\boldsymbol{x})$ of the given instance $\boldsymbol{x} \in \mathbb{R}^d$. Based on the volumes of two instances $\boldsymbol{x}$ and $\widehat{\boldsymbol{x}}$, we can adaptively scale the original distance value $d_{\boldsymbol{\varphi}}(\boldsymbol{x}, \widehat{\boldsymbol{x}})$ to intuitively consider the geometric proximity between the two data balls $\mathcal{B}(\boldsymbol{x}, \mathcal{V}(\boldsymbol{x}))$ and $\mathcal{B}(\widehat{\boldsymbol{x}}, \mathcal{V}(\widehat{\boldsymbol{x}}))$ with their corresponding volumes. As the increased volume makes two data balls closer to each other (see Fig. 2), we naturally use the *negative exponential function* to confine the effect of volume to $(0, 1)$ and thus have the following VAD metric:

$$
\begin{aligned}
\mathcal{D}_{\boldsymbol{\varphi}, \mathcal{V}}&(\boldsymbol{x}, \widehat{\boldsymbol{x}}) \\
&= d_{\boldsymbol{\varphi}}(\boldsymbol{x}, \widehat{\boldsymbol{x}}) \, / \, \mathrm{e}^{\mathcal{V}(\boldsymbol{x}) + \mathcal{V}(\widehat{\boldsymbol{x}})} \\
&= \|\boldsymbol{\varphi}(\boldsymbol{x}) - \boldsymbol{\varphi}(\widehat{\boldsymbol{x}})\|_2 \, / \, \mathrm{e}^{\mathcal{V}(\boldsymbol{x}) + \mathcal{V}(\widehat{\boldsymbol{x}})}, \quad (2)
\end{aligned}
$$

where we can observe that the conventional projected Euclidean distance $d_{\boldsymbol{\varphi}}(\boldsymbol{x}, \widehat{\boldsymbol{x}})$ is *a specific case* of VAD when $\mathcal{V}(\boldsymbol{x}) = \mathcal{V}(\widehat{\boldsymbol{x}}) = 0$. Meanwhile, the significantly large $\mathcal{V}(\boldsymbol{x})$ (or $\mathcal{V}(\widehat{\boldsymbol{x}})$) leads to that $\lim_{\mathcal{V}(\boldsymbol{x}) \to \infty} \mathcal{D}_{\boldsymbol{\varphi}, \mathcal{V}}(\boldsymbol{x}, \widehat{\boldsymbol{x}}) = 0$, in which the large-volume data ball will accommodate the other instances, so that they can share the similar discriminating features with each other. It is easy to validate that such a new VAD metric satisfies the *non-negativity* and *symmetry*, so VAD is always a *semi-metric* for any $\boldsymbol{\varphi}$ and $\mathcal{V}$, and we can obtain that VAD is a *strict metric* if $\mathcal{V}$ is a *constant mapping*. Nevertheless, it is notable that VAD does not satisfy the *triangle inequality* necessarily, because that[1]

$$\lim_{\mathcal{V}(\boldsymbol{x}_2) \to \infty} \frac{\mathcal{D}_{\boldsymbol{\varphi}, \mathcal{V}}(\boldsymbol{x}_1, \boldsymbol{x}_2) + \mathcal{D}_{\boldsymbol{\varphi}, \mathcal{V}}(\boldsymbol{x}_2, \boldsymbol{x}_3)}{\mathcal{D}_{\boldsymbol{\varphi}, \mathcal{V}}(\boldsymbol{x}_1, \boldsymbol{x}_3)} = 0, \quad (3)$$

---

[1]For details, $\lim_{\mathcal{V}(\boldsymbol{x}_2) \to \infty} \frac{\mathcal{D}_{\boldsymbol{\varphi}, \mathcal{V}}(\boldsymbol{x}_1, \boldsymbol{x}_2) + \mathcal{D}_{\boldsymbol{\varphi}, \mathcal{V}}(\boldsymbol{x}_2, \boldsymbol{x}_3)}{\mathcal{D}_{\boldsymbol{\varphi}, \mathcal{V}}(\boldsymbol{x}_1, \boldsymbol{x}_3)} = \lim_{\mathcal{V}(\boldsymbol{x}_2) \to \infty} \frac{\mathrm{e}^{\mathcal{V}(\boldsymbol{x}_3)} \|\boldsymbol{\varphi}(\boldsymbol{x}_1) - \boldsymbol{\varphi}(\boldsymbol{x}_2)\|_2 + \mathrm{e}^{\mathcal{V}(\boldsymbol{x}_1)} \|\boldsymbol{\varphi}(\boldsymbol{x}_2) - \boldsymbol{\varphi}(\boldsymbol{x}_3)\|_2}{\mathrm{e}^{\mathcal{V}(\boldsymbol{x}_2)} \|\boldsymbol{\varphi}(\boldsymbol{x}_1) - \boldsymbol{\varphi}(\boldsymbol{x}_3)\|_2} = \lim_{\mathcal{V}(\boldsymbol{x}_2) \to \infty} C \mathrm{e}^{-\mathcal{V}(\boldsymbol{x}_2)} = 0$, with $C > 0$ independent of $\mathcal{V}(\boldsymbol{x}_2)$.

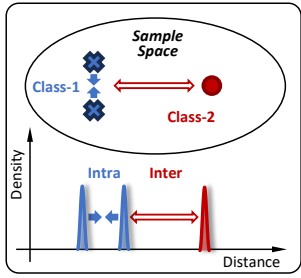 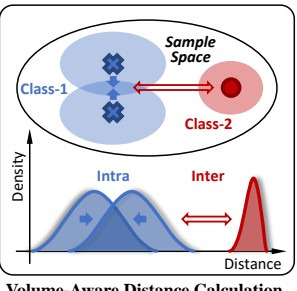

**Volumeless Distance Calculation**     **Volume-Aware Distance Calculation**

*Figure 2.* A comparison between the traditional distance and our proposed VAD. Our metric provides the better coverage to the sample space when calculating distances.

and thus there exists $\widehat{\mathcal{V}}(\cdot)$ such that $(\mathcal{D}_{\boldsymbol{\varphi}, \widehat{\mathcal{V}}}(\boldsymbol{x}_1, \boldsymbol{x}_2) + \mathcal{D}_{\boldsymbol{\varphi}, \widehat{\mathcal{V}}}(\boldsymbol{x}_2, \boldsymbol{x}_3))/\mathcal{D}_{\boldsymbol{\varphi}, \widehat{\mathcal{V}}}(\boldsymbol{x}_1, \boldsymbol{x}_3) < 1$, namely $\mathcal{D}_{\boldsymbol{\varphi}, \widehat{\mathcal{V}}}(\boldsymbol{x}_1, \boldsymbol{x}_2) + \mathcal{D}_{\boldsymbol{\varphi}, \widehat{\mathcal{V}}}(\boldsymbol{x}_2, \boldsymbol{x}_3) < \mathcal{D}_{\boldsymbol{\varphi}, \widehat{\mathcal{V}}}(\boldsymbol{x}_1, \boldsymbol{x}_3)$. This *relaxed triangle property* is friendly to the distance flexibility, because lots of real-world similarity relations do not actually match the triangle inequality necessarily (Fraigniaud et al., 2008; Xu et al., 2022b), and VAD provides a flexible way to describe those relations *as needed*. We also allow *overlaps* among those data balls to make VAD able to reveal similar features belonging to multiple instances.

**Learning with Measure-Head.** As shown in Fig. 3, we introduce an additional network $\mathcal{H}(\cdot)$ ahead of the existing encoder $\boldsymbol{\varphi}(\cdot)$ to measure the volume of the feature embedding result $\boldsymbol{\varphi}(\boldsymbol{x})$, i.e., we let $\mathcal{V} = \mathcal{H} \circ \boldsymbol{\varphi}$ such that

$$\mathcal{V}(\boldsymbol{x}) = \mathcal{H}[\boldsymbol{\varphi}(\boldsymbol{x})] = \mathrm{ReLU}[\boldsymbol{\alpha}^\top \mathrm{ReLU}(\boldsymbol{W} \cdot \boldsymbol{\varphi}(\boldsymbol{x}))], \quad (4)$$

where the measure-head $\mathcal{H} : \mathbb{R}^m \to \mathbb{R}^+$ is implemented with a classical *multi-layer perceptron* (MLP) (Hastie, 2009). Here $\boldsymbol{W} \in \mathbb{R}^{k \times m}$, $\boldsymbol{\alpha} \in \mathbb{R}^k$, and $k$ is the dimensionality of hidden layer. In this way, the volume prediction effectively guides the training of the encoder network to capture useful features. Meanwhile, the reliable features extracted by the encoder $\boldsymbol{\varphi}$ can also assist the measure-head $\mathcal{H}$ in the faithful volume determination. Both the encoder $\boldsymbol{\varphi}$ and the measure-head $\mathcal{H}$ are learned with the conventional empirical loss. For the widely used $(n + 1)$-*tuplet/Npair loss* in metric learning (Sohn, 2016) and the *NCE loss* (Chen et al., 2020) in contrastive learning, based on the given training set $\mathcal{X} = \{\boldsymbol{x}_i \in \mathbb{R}^d | i = 1, 2, \ldots, N\}$, our corresponding empirical risk can be easily summarized as

$$
\begin{aligned}
\mathcal{L}_{\mathrm{emp}}&(\boldsymbol{\varphi}, \mathcal{H}) \\
&= \mathbb{E}_{\boldsymbol{x}, \{b_j\}_{j=1}^n}[\ell(\boldsymbol{\varphi}, \mathcal{H}; \{\boldsymbol{x}, \boldsymbol{x}_{b_1}, \boldsymbol{x}_{b_2}, \ldots, \boldsymbol{x}_{b_n}\})] \\
&= \mathbb{E}_{\boldsymbol{x}, \{b_j\}_{j=1}^n}\left[-\log \frac{\mathrm{e}^{-\mathcal{D}_{\boldsymbol{\varphi}, \mathcal{V}}(\boldsymbol{x}, \boldsymbol{x}^+)/\gamma}}{\mathrm{e}^{-\mathcal{D}_{\boldsymbol{\varphi}, \mathcal{V}}(\boldsymbol{x}, \boldsymbol{x}^+)/\gamma} + \sum_{j=1}^n \mathrm{e}^{-\mathcal{D}_{\boldsymbol{\varphi}, \mathcal{V}}(\boldsymbol{x}, \boldsymbol{x}_{b_j})/\gamma}}\right],
\end{aligned}
$$
$$(5)$$

where $\gamma > 0$ is a temperature parameter, and the anchor instance $\boldsymbol{x}$ is sampled from $\mathcal{X}$. Here $\boldsymbol{x}^+$ is directly obtained

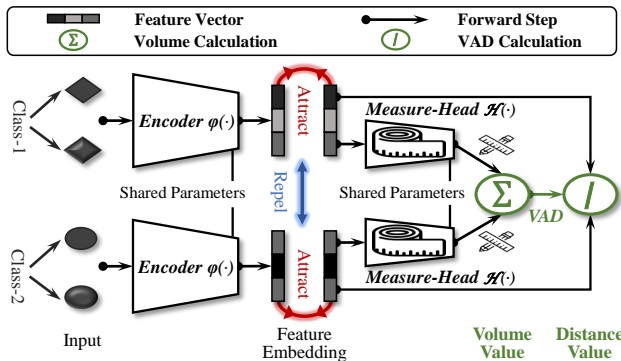

*Figure 3.* The overall framework of our proposed volume-aware distance based similarity learning.

from the perturbed $\boldsymbol{x}$ for the unsupervised (contrastive learning) case, or it is chosen randomly from the *intra-class* set $\mathscr{X}^{+}(\boldsymbol{x}) = \{\boldsymbol{z} \,|\, y_{\boldsymbol{z}} = y_{\boldsymbol{x}}, \boldsymbol{z} \in \mathscr{X} \backslash \{\boldsymbol{x}\}\}$ for the supervised (metric learning) case. Correspondingly, the mini-batch instances $\{\boldsymbol{x}_{b_1}, \boldsymbol{x}_{b_2}, \ldots, \boldsymbol{x}_{b_n}\}$ are directly selected from $\mathscr{X} \backslash \{\boldsymbol{x}\}$ for the unsupervised case, or from the *inter-class* set $\mathscr{X}^{-}(\boldsymbol{x}) = \{\boldsymbol{z} \,|\, y_{\boldsymbol{z}} \neq y_{\boldsymbol{x}}, \boldsymbol{z} \in \mathscr{X}\}$ for the supervised case ($y_{\boldsymbol{x}}$ is the class label of $\boldsymbol{x}$).

**Volume Expansion Regularizer.** Minimizing the above empirical loss can finally learn a VAD metric which provides the consistent prediction with the supervisory information. However, we further need a new regularization term to explicitly encourage the volume determination of instances towards a generalizable learning result. Actually, we want the data points in the embedding space to capture/cover their intrinsically similar features as much as possible, so here we simply encourage the volume of each instance $\mathcal{V}(\boldsymbol{x})$ to be *as large as possible*. Such a volume expansion strategy makes the data balls $\mathcal{B}(\boldsymbol{x}_1, \mathcal{V}(\boldsymbol{x}_1)), \mathcal{B}(\boldsymbol{x}_2, \mathcal{V}(\boldsymbol{x}_2)), \ldots, \mathcal{B}(\boldsymbol{x}_N, \mathcal{V}(\boldsymbol{x}_N))$ to accommodate each other as many as they can, and thus we propose the following *volume expansion regularizer* (VER):

$$\mathcal{R}_{\text{expand}}(\boldsymbol{\varphi}, \mathcal{H}) = \mathbb{E}_{\{b_j\}_{j=1}^n} \left[ \sum_{j=1}^n e^{-\mathcal{V}(\boldsymbol{x}_{b_j})} \right], \quad (6)$$

where the mini-batch instances $\{\boldsymbol{x}_{b_1}, \boldsymbol{x}_{b_2}, \ldots, \boldsymbol{x}_{b_n}\}$ are directly selected from $\mathscr{X}$, and the natural exponential function is employed again for numerical simplicity.

### 3.2. Optimization & Convergence

Here we elaborate on the optimization process and convergence property of our learning algorithm.

**Stochastic Gradient & Boundness.** By combining the above empirical loss in Eq. (5) and VER in Eq. (6), we finally obtain the objective of our *volume-aware distance based similarity learning* (VADSL):

$$\min_{\boldsymbol{\varphi}, \mathcal{H}} \{ \mathcal{F}(\boldsymbol{\varphi}, \mathcal{H}) = \mathcal{L}_{\text{emp}}(\boldsymbol{\varphi}, \mathcal{H}) + \lambda \mathcal{R}_{\text{expand}}(\boldsymbol{\varphi}, \mathcal{H}) \}, \quad (7)$$

**Algorithm 1** Solving Eq. (7) via SGD.

**Input:** training set $\mathscr{X} = \{\boldsymbol{x}_i\}_{i=1}^N$; step size $\eta > 0$; regularization parameter $\lambda > 0$; batch size $n \in \mathbb{N}_+$; randomly initialized $\boldsymbol{\varphi}^{(0)}$; maximum iteration number $T$.

**For** $t$ **from** 1 **to** $T$:

1). Uniformly pick $(n+1)$ instances $\{\boldsymbol{x}_{b_j}\}_{j=0}^n$ from $\mathscr{X}$;

2). Compute $\nabla_{\mathcal{H}} \mathcal{L}_{\text{emp}}(\{b_j\}_{j=1}^n)$, $\nabla_{\mathcal{H}} \mathcal{R}_{\text{expand}}(\{b_j\}_{j=1}^n)$, $\nabla_{\boldsymbol{\varphi}} \mathcal{L}_{\text{emp}}(\{b_j\}_{j=1}^n)$ and $\nabla_{\boldsymbol{\varphi}} \mathcal{R}_{\text{expand}}(\{b_j\}_{j=1}^n)$ according to Eq. (8) and Eq. (9);

3). Update the learning parameter:

$$\begin{cases} \boldsymbol{\varphi}^{(t)} = \boldsymbol{\varphi}^{(t-1)} - \eta(\nabla_{\boldsymbol{\varphi}} \mathcal{L}_{\text{emp}} + \lambda \nabla_{\boldsymbol{\varphi}} \mathcal{R}_{\text{expand}}); \\ \mathcal{H}^{(t)} = \mathcal{H}^{(t-1)} - \eta(\nabla_{\mathcal{H}} \mathcal{L}_{\text{emp}} + \lambda \nabla_{\mathcal{H}} \mathcal{R}_{\text{expand}}); \end{cases} (10)$$

**End**

**Output:** converged $\boldsymbol{\varphi}^{(T)}$ and $\mathcal{H}^{(T)}$.

where $\lambda > 0$ is a trade-off parameter. Then we are able to optimize our learning objective Eq. (7) in a stochastic way, where we only need to specify the stochastic terms of both $\mathcal{L}_{\text{emp}}(\boldsymbol{\varphi}, \mathcal{H})$ and $\mathcal{R}_{\text{expand}}(\boldsymbol{\varphi}, \mathcal{H})$ for a given mini-batch. It is worth pointing out that the learning parameters $\boldsymbol{\varphi}$ and $\mathcal{H}$ are interdependent, where the stochastic gradient of $\mathcal{R}_{\text{expand}}$ w.r.t. $\mathcal{H}$ can be calculated as

$$\nabla_{\mathcal{H}} \mathcal{R}_{\text{expand}}(\{b_j\}_{j=1}^n) = \sum_{j=1}^n -e^{-\mathcal{V}(\boldsymbol{x}_{b_j})} \frac{\partial \mathcal{V}(\boldsymbol{x}_{b_j})}{\partial \mathcal{H}}, \quad (8)$$

and the stochastic gradient $\nabla_{\mathcal{H}} \mathcal{L}_{\text{emp}}(\{b_j\}_{j=1}^n)$ is a direct calculation of $\partial \ell / \partial \mathcal{H}$. Based on the gradient result w.r.t. $\mathcal{H}$, we can further obtain that

$$\begin{cases} \nabla_{\boldsymbol{\varphi}} \mathcal{L}_{\text{emp}}(\{b_j\}_{j=1}^n) = \nabla_{\mathcal{H}} \mathcal{L}_{\text{emp}}(\{b_j\}_{j=1}^n) \cdot \frac{\partial \mathcal{H}}{\partial \boldsymbol{\varphi}}, \\ \nabla_{\boldsymbol{\varphi}} \mathcal{R}_{\text{expand}}(\{b_j\}_{j=1}^n) = \nabla_{\mathcal{H}} \mathcal{R}_{\text{expand}}(\{b_j\}_{j=1}^n) \cdot \frac{\partial \mathcal{H}}{\partial \boldsymbol{\varphi}}, \end{cases} (9)$$

which only requires a single derivative computation of $\partial \mathcal{H} / \partial \boldsymbol{\varphi}$ to get the gradient w.r.t. $\boldsymbol{\varphi}$. The detailed iteration steps based on *stochastic gradient descent* (SGD) (Reddi et al., 2016) are summarized in Algorithm 1. Existing work reveals that the convergence of iteration points $(\boldsymbol{\varphi}^{(1)}, \mathcal{H}^{(1)}), (\boldsymbol{\varphi}^{(2)}, \mathcal{H}^{(2)}), \ldots, (\boldsymbol{\varphi}^{(T)}, \mathcal{H}^{(T)})$ can naturally inherit from SGD as long as the objective function $\mathcal{F}(\boldsymbol{\varphi}, \mathcal{H})$ is *Lipschitz-smooth* and *gradient-bounded* (Huang et al., 2019). We have the following theorem to ensure the gradient boundness and Lipschitz smoothness for our learning objective $\mathcal{F}(\boldsymbol{\varphi}, \mathcal{H})$ in Eq. (7).

**Theorem 1.** *The learning objective $\mathcal{F}(\boldsymbol{\varphi}, \mathcal{H})$ is always gradient-bounded and Lipschitz-smooth if the encoder $\boldsymbol{\varphi}(\cdot)$ is gradient-bounded and Lipschitz-smooth.*

This implies that the gradient boundness and Lipschitz smoothness of our learning objective are directly inherited from the original encoder network $\boldsymbol{\varphi}$ and well preserved in the VADSL. As a result, the practical convergence of our learning algorithm is theoretically guaranteed.

## 4. Theoretical Analyses

In this section, we provide in-depth theoretical results to investigate the generalization ability, the distance flexibility, and the sample-space coverage of our proposed method.

### 4.1. Generalization Error Bound

Here we would like to prove that our algorithm provides a tighter *generalization error bound* (GEB) (Chen et al., 2021) compared with conventional similarity learning approaches. This is achieved by analyzing the convergence rate of the GEB with respect to the sample size $N$ and by demonstrating how the proposed regularization term $\mathcal{R}_{\text{expand}}$ contributes to tightening this bound. Specifically, for the underlying data distribution $\mathscr{D}$, we denote the expected risk $\widetilde{\mathcal{L}}_{\text{emp}}(\boldsymbol{\varphi}, \mathcal{H}; \mathscr{D}) = \mathbb{E}_{\{\boldsymbol{t}_i | \boldsymbol{t}_i \sim \mathscr{D}\}_{i=1}^N}[\mathcal{L}_{\text{emp}}(\boldsymbol{\varphi}, \mathcal{H}; \{\boldsymbol{t}_i\}_{i=1}^N)]$ and discuss how far it is from the empirical risk $\mathcal{L}_{\text{emp}}(\boldsymbol{\varphi}, \mathcal{H})$.

**Theorem 2.** *For any $\boldsymbol{\varphi}$ learned from the objective $\mathcal{F}(\boldsymbol{\varphi}, \mathcal{H})$ and any given constant $\delta \in (0, 1)$, we have that with probability at least $1 - \delta$,*

$$\begin{aligned} &|\mathcal{L}_{\text{emp}}(\boldsymbol{\varphi}, \mathcal{H}) - \widetilde{\mathcal{L}}_{\text{emp}}(\boldsymbol{\varphi}, \mathcal{H}; \mathscr{D})| \\ &\leq \omega(n)\log(1 + \mathcal{D}_{\max})\sqrt{[\ln(2/\delta)]/(2N\theta(\lambda))}, \end{aligned} \quad (11)$$

*where $\mathcal{D}_{\max} = \max\{\mathcal{D}_{\boldsymbol{\varphi}, \mathcal{H}}(\boldsymbol{t}, \widehat{\boldsymbol{t}}) | \boldsymbol{t}, \widehat{\boldsymbol{t}} \in \mathscr{X}\}$. The function $\omega(n) = \log\left(e^2/n + 1\right)$ is monotonically decreasing w.r.t. $n$. The function $\theta(\lambda) = C\lambda$ where the positive constant $C$ is independent of $\boldsymbol{\varphi}$, $\mathcal{H}$, and $\mathscr{X}$.*

The error bound in Eq. (11) is dominated by two factors. First, the generalization error bound decreases as the sample size $N$ and batch size $n$ increase. This behavior aligns with observations in prior research, where larger datasets and batch sizes typically lead to reduced generalization error. More importantly, the error bound tightens as the regularization parameter $\lambda$ increases. This is because $\theta(\lambda)$, defined as a monotonically increasing function of $\lambda$, grows larger, thereby enabling faster convergence of the empirical risk $\mathcal{L}_{\text{emp}}(\boldsymbol{\varphi}, \mathcal{H})$ to the expected risk $\widetilde{\mathcal{L}}_{\text{emp}}(\boldsymbol{\varphi}, \mathcal{H}; \mathscr{D})$. It means that our VER regularizer $\mathcal{R}_{\text{expand}}$ can accelerate the empirical risk convergence to the expected risk.

### 4.2. Distance Flexibility of VAD

Now we further analyze the flexibility of our VAD in ordering distance sequence. As the most crucial thing in similarity learning is the *relative distance* (namely the *order* of distance sequence) but not the absolute distance itself (Liu & Tsang, 2015; Ge, 2018), we want to show that for the given embedding $\boldsymbol{\varphi}$, our VAD always has a volume measure $\mathcal{V}$ to *fit any predefined distance sequence*. Specifically, we have the following theorem to reveal the flexibility of our VAD.

**Theorem 3.** *For the given dataset $\mathscr{X} = \{\boldsymbol{x}_i \in \mathbb{R}^d | i = 1, 2, \ldots, N\}$, feature embedding $\boldsymbol{\varphi}$, and partial ordering*

$(a_1, b_1) \succcurlyeq (a_2, b_2) \succcurlyeq \cdots \succcurlyeq (a_{\text{C}_N^2}, b_{\text{C}_N^2})$, *there always exists $\mathcal{H}^* : \mathbb{R}^m \to \mathbb{R}^+$ such that*

$$\begin{aligned} \mathcal{D}_{\boldsymbol{\varphi}, \mathcal{V}^*}(\boldsymbol{x}_{a_1}, \boldsymbol{x}_{b_1}) &\geq \mathcal{D}_{\boldsymbol{\varphi}, \mathcal{V}^*}(\boldsymbol{x}_{a_2}, \boldsymbol{x}_{b_2}) \geq \cdots \\ &\geq \mathcal{D}_{\boldsymbol{\varphi}, \mathcal{V}^*}(\boldsymbol{x}_{\text{C}_N^2}, \boldsymbol{x}_{\text{C}_N^2}), \end{aligned} \quad (12)$$

*where $\bigcup_{i=1}^{\text{C}_N^2}\{(a_i, b_i)\} = \{(A, B) | A < B, \text{ where } A, B = 1, 2, \ldots, N\}$, $\mathcal{V}^* = \mathcal{H}^* \circ \boldsymbol{\varphi}$, and $\boldsymbol{\varphi}$ is independent of $\mathcal{H}^*$.*

The above interesting result implies that our VAD has a flexible adaptability even if the feature embedding $\boldsymbol{\varphi}$ is fixed. The measure-head can always learn an optimal $\mathcal{H}^*$ to make the finally predicted distances satisfy any given *partial ordering relations*. This will also be beneficial to the feature extraction of the encoder $\boldsymbol{\varphi}$ because the measure-head $\mathcal{H}^*$ successfully *separates the distance measurement apart from the feature embedding* and explicitly keeps it as a white-box module in the overall learning framework.

### 4.3. Finite Covering of Volume-Awareness

As our basic motivation, volume-awareness is introduced to extend the traditional data points $\boldsymbol{x}_1, \boldsymbol{x}_2, \ldots, \boldsymbol{x}_N$ to the data balls $\mathcal{B}(\boldsymbol{x}_1, \mathcal{V}(\boldsymbol{x}_1)), \mathcal{B}(\boldsymbol{x}_2, \mathcal{V}(\boldsymbol{x}_2)), \ldots, \mathcal{B}(\boldsymbol{x}_N, \mathcal{V}(\boldsymbol{x}_N))$, so that the whole sample space can be covered as much as possible. Therefore, here we investigate if it is possible to cover the sample space with any given coverage ratio.

**Theorem 4.** *Suppose that $\boldsymbol{\varphi}$ is feature-normalized such that $\boldsymbol{z} = \boldsymbol{\varphi}(\boldsymbol{x}) \in [L, U]^m, \forall \boldsymbol{x} \in \mathbb{R}^d$. Then for any given $\rho \in (0, 1)$, there exists sufficiently large $N$ such that*

$$\frac{\int_{\boldsymbol{z} \in [L, U]^m} \text{sign}[\boldsymbol{z} \in \bigcup_{i=1}^N \mathcal{B}(\boldsymbol{x}_i, \mathcal{V}(\boldsymbol{x}_i))]\mathrm{d}\boldsymbol{z}}{\int_{\boldsymbol{z} \in [L, U]^m} 1\mathrm{d}\boldsymbol{z}} \geq \rho, \quad (13)$$

*where the integral values $\int_{\boldsymbol{z} \in [L, U]^m} \text{sign}[\boldsymbol{z} \in \bigcup_{i=1}^N \mathcal{B}(\boldsymbol{x}_i, \mathcal{V}(\boldsymbol{x}_i))]\mathrm{d}\boldsymbol{z} = \int_{\boldsymbol{z} \in \bigcup_{i=1}^N \mathcal{B}(\boldsymbol{x}_i, \mathcal{V}(\boldsymbol{x}_i))} \mathrm{d}\boldsymbol{z}$ and $\int_{\boldsymbol{z} \in [L, U]^m} 1\mathrm{d}\boldsymbol{z}$ are the volumes of all data balls and the whole feature space $[L, U]^m$, respectively.*

This theorem clearly answers that the volume-awareness can sufficiently cover (with a desired coverage ratio of $\rho$) the whole sample space by using only the finite number of (namely $N$) instances. Intuitively, this is because each instance has been endowed with a volume (i.e., the data ball), and thereby it becomes easy to employ those volume-specified entities to fill in the sample space.

## 5. Experimental Results

We conduct experiments to evaluate the performance of our proposed method using real-world datasets. We first conduct ablation studies to reveal the usefulness of our newly introduced block/regularizer. Then we compare our proposed learning algorithm with existing state-of-the-art models in both the supervised metric learning and unsupervised

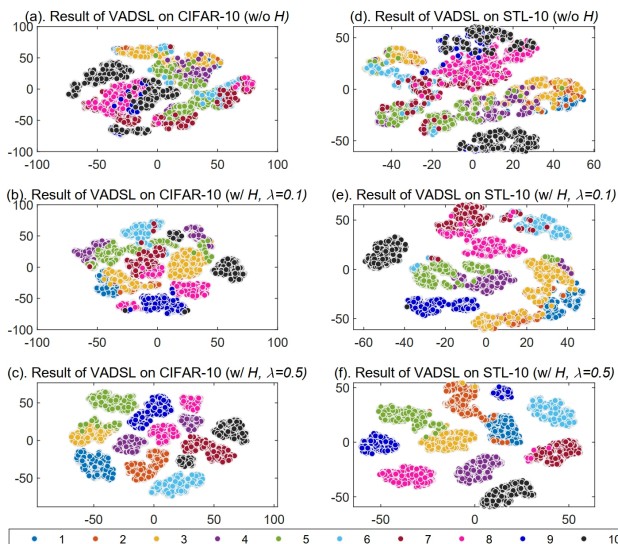

Figure 4. The t-SNE visualizations of our VADSL on CIFAR-10 and STL-10 datasets.

contrastive learning tasks. Both the training and test processes are implemented on Pytorch (Paszke et al., 2019) with TeslaV100 GPUs, where the regularization parameter $\lambda$ is set to 0.5. The dimensionality $m$ and the parameter $\gamma$ in Eq. (5) are set to 512 and 0.2, respectively. We use a 128-dimensional hidden layer for our $\mathcal{H}(\cdot)$ in Eq. (4). The hyper-parameters of compared methods are set to the recommended values according to their original papers.

### 5.1. Ablation Studies & Visualization Results

In this subsection, we explore the effectiveness of our proposed VAD and its associated regularizer on different tasks.

For the supervised task, we adopt different feature encoders (*BN-Inception* (Ioffe & Szegedy, 2015) for *Npair* (Sohn, 2016), and *ResNet-50* (He et al., 2016) for *ProxyAnchor* (Kim et al., 2020) and *MetricFormer* (Yan et al., 2022b)) to assess the performance of our method in metric learning. The results are presented in Tab. 1, where we record the test accuracy of all compared methods on *CAR-196* (Krause et al., 2013) and *CUB-200* (Welinder et al., 2010) datasets (with 500 epochs, learning rate = $10^{-3}$, and batch size = 512). We observe that our VADSL performs well across all three baseline methods, providing stable performance in various scenarios with different embedding sizes. Our approach consistently improves upon all baseline methods when equipped with the measure-head $\mathcal{H}$. Furthermore, increasing the regularization parameter $\lambda$ from 0.1 to 0.5 leads to noticeable improvements in recognition accuracy, highlighting the critical importance of our regularization term, the volume expansion.

For the unsupervised task, here we adopt the ResNet-50

Table 1. Classification accuracy rates of baseline methods and our method on CAR-196 and CUB-200 datasets (feature embedding sizes are 128 and 512).

| METHOD | CAR-196 | | | | CUB-200 | | | |
|---|---|---|---|---|---|---|---|---|
| | 128-dim. | | 512-dim. | | 128-dim. | | 512-dim. | |
| | R@1 | R@8 | R@1 | R@8 | R@1 | R@8 | R@1 | R@8 |
| Npair(BN) w/o $\mathcal{H}$ | 68.36 | 86.01 | 82.37 | 95.12 | 58.12 | 78.72 | 65.38 | 90.82 |
| VADSL[N. w/ $\mathcal{H}$ ($\lambda=0$)] | 68.36 | 86.21↑ | 82.48↑ | 95.29↑ | 58.56↑ | 79.12↑ | 65.88↑ | 91.22↑ |
| VADSL[N. w/ $\mathcal{H}$ ($\lambda=0.1$)] | 68.39↑ | 88.26↑ | 85.32↑ | 96.12↑ | 58.14↑ | 80.34↑ | 66.32↑ | 91.87↑ |
| VADSL[N. w/ $\mathcal{H}$ ($\lambda=0.5$)] | **70.23↑** | **90.39↑** | **89.26↑** | **96.32↑** | **62.07↑** | **82.35↑** | **69.07↑** | **92.55↑** |
| ProxA.(R50) w/o $\mathcal{H}$ | 69.24 | 87.86 | 87.71 | 97.86 | 62.12 | 79.26 | 69.72 | 92.41 |
| VADSL[P. w/ $\mathcal{H}$ ($\lambda=0$)] | 69.27↑ | 87.86 | 87.77↑ | 97.95↑ | 62.12 | 79.86↑ | 69.91↑ | 92.69↑ |
| VADSL[P. w/ $\mathcal{H}$ ($\lambda=0.1$)] | 69.22 | 88.84↑ | 89.85↑ | 98.26↑ | 62.26↑ | 80.83↑ | 71.25↑ | 92.69↑ |
| VADSL[P. w/ $\mathcal{H}$ ($\lambda=0.5$)] | **70.26↑** | **91.32↑** | **92.42↑** | **98.85↑** | **63.19↑** | **82.21↑** | **73.92↑** | **94.14↑** |
| M.F.(R50) w/o $\mathcal{H}$ | 72.42 | 89.53 | 91.76 | 97.21 | 69.33 | 85.12 | 74.42 | 92.53 |
| VADSL[M. w/ $\mathcal{H}$ ($\lambda=0$)] | 72.42 | 89.53 | 91.66 | 97.21 | 69.77↑ | 85.82↑ | 74.42 | 92.63↑ |
| VADSL[M. w/ $\mathcal{H}$($\lambda=0.1$)] | 73.45↑ | 91.26↑ | 91.48 | 97.77↑ | 70.95↑ | 86.48↑ | 74.72↑ | 92.65↑ |
| VADSL[M. w/ $\mathcal{H}$ ($\lambda=0.5$)] | **73.59↑** | **92.35↑** | **92.23↑** | **98.45↑** | **72.55↑** | **88.35↑** | **75.74↑** | **93.45↑** |

backbone for several contrastive learning methods including *SimCLR* (Chen et al., 2020), *BYOL* (Grill et al., 2020), and *HCL* (Robinson et al., 2021). In Fig. 5, we record the classification accuracy of all compared methods on *CIFAR-10* (Krizhevsky et al., 2009) and *STL-10* datasets (Coates et al., 2011), where we can observe that our method consistently improves the corresponding baseline results in all scenarios. To be more intuitive, we also conduct the *t-SNE* embedding (Van der Maaten & Hinton, 2008) to obtain the 2-dimensional data points to better understand the usefulness of our introduced new component. In Fig. 4, VADSL (w/ measure-head $\mathcal{H}$) can successfully obtain the better separability than the baseline result (w/o $\mathcal{H}$), where the results of $\lambda = 0.5$ achieve very satisfactory separability. These results clearly demonstrate the crucial role of maintaining the measure-head network $\mathcal{H}$ along with the corresponding regularizer $\mathcal{R}_{\text{expand}}$ in our approach.

### 5.2. Experiments on Supervised Metric Learning

In this subsection, we evaluate the effectiveness and superiority of VADSL on the supervised task of metric learning.

**Image Data.** We assess the performance of VADSL in general image retrieval tasks across datasets on *CAR*-196 (Krause et al., 2013), *CUB*-200 (Welinder et al., 2010), *SOP* (Oh Song et al., 2016), and *In-Shop* (Liu et al., 2016). The methods we compare include *JDR* (Chu et al., 2020), *IBC* (Seidenschwarz et al., 2021), *ContextSimilarity* (Liao et al., 2023), *MetricFormer* (Yan et al., 2022b), *MFC* (Furusawa, 2024), and *DASL* (Chen et al., 2024a). All compared methods are incorporated into the ResNet-50 backbone, and we refer to the combinations of our approach with Npair loss and ProxyAnchor loss as VADSL-NP and VADSL-PA, respectively. The NMI and Recall@R scores of all methods are shown in Tab. 2, where we clearly observe that DASL, MFC, and our methods obtain higher accuracies than other methods. Compared with those strong baseline methods, our VADSL can further achieve either better or competitive NMI and Recall@R scores.

*Table 2.* Performance of all compared methods (with ResNet-50 backbone) on *CAR-196, CUB-200, SOP,* and *In-Shop* datasets. The best two results are **bolded** and underlined, respectively.

| METHOD | CAR-196 | | | | CUB-200 | | | | SOP | | | | In-Shop | | | |
|---|---|---|---|---|---|---|---|---|---|---|---|---|---|---|---|---|
| | NMI | R@1 | R@4 | R@8 | NMI | R@1 | R@4 | R@8 | NMI | R@1 | R@10 | R@100 | NMI | R@1 | R@10 | R@20 |
| Npair(Sohn, 2016) | 69.50 | 82.57 | 94.97 | 95.92 | 69.53 | 64.52 | 85.63 | 91.15 | 91.11 | 76.21 | 88.43 | 92.08 | 85.12 | 90.21 | 98.01 | 98.72 |
| ProxyA.(Kim et al., 2020) | 75.72 | 87.71 | 95.76 | 97.86 | 72.31 | 69.72 | 87.01 | 92.41 | 91.02 | 78.39 | 90.48 | 96.16 | 90.22 | 91.49 | 98.12 | 98.83 |
| JDR(Chu et al., 2020) | 70.56 | 84.86 | 94.56 | 97.21 | 70.32 | 69.44 | 87.01 | 91.33 | 92.21 | 79.21 | 90.53 | 96.01 | 91.69 | 92.11 | 97.63 | 98.31 |
| IBC(Seidenschwarz et al., 2021) | 74.82 | 88.11 | 96.21 | 98.21 | 74.01 | 70.32 | 87.61 | 92.72 | 92.61 | 81.42 | 91.32 | 95.89 | 91.34 | 92.82 | 98.52 | 99.13 |
| MetricF.(Yan et al., 2022b) | 76.23 | 91.76 | 96.31 | 97.21 | 75.41 | 74.42 | 85.75 | 92.53 | 92.71 | 82.23 | 92.62 | 96.33 | 89.32 | 91.25 | 97.82 | 98.36 |
| ContextS.(Liao et al., 2023) | 76.32 | 91.80 | 97.14 | 98.41 | 74.01 | 71.91 | 88.82 | 93.42 | 92.61 | 82.63 | 92.56 | 96.74 | 86.89 | 90.73 | 97.82 | 98.51 |
| MFC(Furusawa, 2024) | 75.96 | 91.61 | 97.64 | 98.57 | 73.37 | 71.83 | 88.25 | 93.27 | 91.37 | 79.59 | 92.36 | 96.57 | 91.78 | 92.78 | 98.86 | 98.87 |
| DASL(Chen et al., 2024a) | 77.32 | 92.31 | 97.82 | 98.90 | 76.50 | 73.96 | 90.54 | 94.21 | 93.86 | 83.32 | 93.86 | 97.95 | 86.32 | 90.62 | 97.56 | 98.95 |
| VADSL-NP (ours) | 76.86↑ | 93.42↑ | 97.65↑ | 98.97↑ | 76.82↑ | 77.36↑ | 90.62↑ | 94.61↑ | 93.85↑ | 84.55↑ | 94.12↑ | 98.92↑ | 92.38↑ | 93.38↑ | 98.21↑ | **99.53↑** |
| VADSL-PA (ours) | **78.22↑** | **94.13↑** | **98.28↑** | **99.40↑** | **77.55↑** | **78.69↑** | **91.45↑** | **95.12↑** | **94.96↑** | **86.23↑** | **95.36↑** | **98.92↑** | **93.15↑** | **93.82↑** | **98.95↑** | 99.28↑ |

*Table 3.* Accuracy rates of all compared methods on AgeDB30, CFPFP, and MegaFace datasets.

| METHOD | Face Verification | | Face Identif. (MegaFace) | | |
|---|---|---|---|---|---|
| | Age. | CFP. | M.-$10^6$ | M.-$10^5$ | M.-$10^4$ |
| Softmax | 91.30 | 93.39 | 80.43 | 87.11 | 92.83 |
| Sph.+$\ell_2$(Liu et al., 2017) | 93.42 | 94.30 | 88.38 | 92.86 | 95.93 |
| Sph.+SEC(Zhang et al., 2020) | 93.45 | 94.39 | 88.42 | 92.79 | 95.88 |
| Arc.+$\ell_2$(Deng et al., 2019) | 93.93 | 94.77 | 90.68 | 94.34 | 96.83 |
| Arc.+SEC(Zhang et al., 2020) | 93.82 | 94.91 | 90.91 | 94.56 | 96.95 |
| MFC(Furusawa, 2024) | 94.98 | 95.21 | 90.82 | 95.68 | 97.32 |
| VADSL (ours, Sph.+VAD) | 94.92↑ | 95.37↑ | 89.75↑ | 93.77↑ | 96.42↑ |
| VADSL (ours, Arc.+VAD) | **95.21↑** | **96.24↑** | **91.78↑** | **95.89↑** | **97.45↑** |

**Facial Data.** We employ *CASIA-WebFace* (Yi et al., 2014) as the training set while using *AgeDB30* (Moschoglou et al., 2017), *CFP-FP* (Sengupta et al., 2016), and *MegaFace* (Kemelmacher-Shlizerman et al., 2016) as the test sets. For all methods, we set the batch size to 256 and the embedding size to 512, using the ResNet-50 backbone. The methods compared include various regularized versions of *Sphereface* (Zhang et al., 2020) and *Arcface* (Deng et al., 2019), and the deep metric learning loss *MFC* (Furusawa, 2024). As shown in Tab. 3, we can clearly observe that our VAD enhances the performance of both Sphereface and Arcface in all cases. For example, on MegaFace with $10^6$ distractors, the accuracies of Sphereface and Arcface are boosted by 1.37% and 1.1%, respectively.

### 5.3. Experiments on Unsupervised Contrastive Learning

In this subsection, we use different domains of data to evaluate the effectiveness and superiority of VADSL on the unsupervised task of contrastive learning.

**Image Data.** We employ ResNet-50 as the backbone and integrate our method with *SimCLR* (Chen et al., 2020) and *SwAV* (Caron et al., 2020), yielding the results labeled as VADSL (cluster-free) and VADSL (cluster-used), respectively. We train our method on *ImageNet-100* and *ImageNet-1K* (Russakovsky et al., 2015), and compare it with existing representative approaches including *HCL* (Robinson et al., 2021), *PCL* (Li et al., 2021), *BYOL* (Grill et al., 2020), *GCA* (Chen et al., 2024b), and *INTL* (Weng et al., 2024). Additionally, we implement our method using the popular *ViT-B/16*

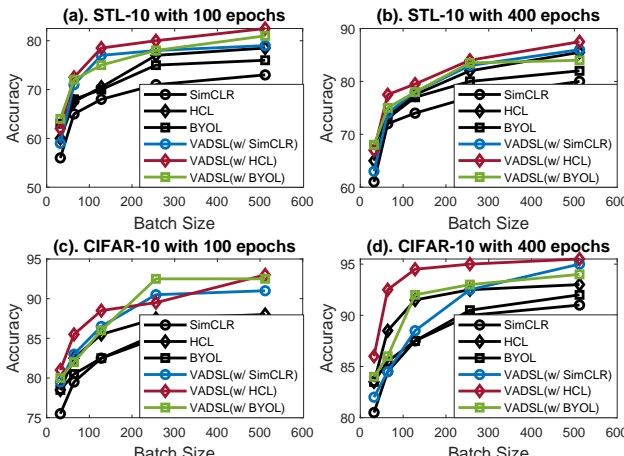

*Figure 5.* Classification accuracy of all methods on STL-10 and CIFAR-10 datasets with different training epochs, where the (negative) batch size is from 32 to 512.

backbone and compare it with three more methods including *DINO* (Caron et al., 2021), *iBOT* (Zhou et al., 2022), and *MTE* (Li et al., 2024). The classification accuracy is evaluated using the linear softmax (i.e., the Top-1 score and Top-5 score of *linear probing*) and the $k$-NN classification (here $k = 8$). From the results shown in Tab. 4, it is evident that our method consistently improves both SimCLR and SwAV by at least 3% in most cases. When leveraging the powerful ViT-B/16 encoder, our method consistently improves the baselines and surpasses three state-of-the-art methods (DINO, iBOT, and MTE) across multiple datasets.

**Text Data.** In this experiment, we use the *STS* dataset (Agirre et al., 2016) (including the tasks of *STS12, STS13, STS14, STS15,* and *STS16*). Following the approach in *Sim-CSE* (Gao et al., 2021), we utilize pre-trained *BERT* (Devlin et al., 2018) checkpoints and compare our method with *InforMin-CL* (Chen et al., 2022b), *misCSE* (Klein & Nabi, 2022), *PCL* (Li et al., 2021), *SCL* (Wu et al., 2022b), and *AD-NCE* (Wu et al., 2024). As we can observe from Tab. 5, our VADSL obtains considerable improvements on the baseline method SimCSE. Meanwhile, our method can outperform the representative methods misCSE and InforMin-CL in

*Table 4.* Classification accuracy (%) of all methods on ImageNet-100 and ImageNet-1K datasets. The batch sizes are set to 1024 and 512 for ResNet-50 and ViT-B/16 backbones, respectively. Here the best and second-best results are **bolded** and underlined, respectively.

| METHOD | FC100 | | Plant | | ImageNet-100 | | | | | | | | | ImageNet-1K | | | | | | #Arch. /#Total-Param. |
|---|---|---|---|---|---|---|---|---|---|---|---|---|---|---|---|---|---|---|---|---|
| | 5-W 1-S | 5-W 5-S | 5-W 1-S | 5-W 5-S | 100 epochs | | | 400 epochs | | | 300 epochs | | | 800 epochs | | | | | | |
| | | | | | $k$-NN | Top-1 | Top-5 | $k$-NN | Top-1 | Top-5 | $k$-NN | Top-1 | Top-5 | $k$-NN | Top-1 | Top-5 | | | | |
| SimCLR(Chen et al., 2020) | 38.45 | 54.93 | 67.29 | 85.29 | 55.9 | 61.3 | 78.6 | 70.6 | 75.2 | 92.1 | 64.2 | 67.4 | 87.9 | 66.1 | 69.3 | 89.6 | | | | Res.50 / 23MB |
| BYOL(Grill et al., 2020) | 35.49 | 55.91 | 65.92 | 85.91 | 56.3 | 65.5 | 77.8 | 69.2 | 73.2 | 90.1 | 66.9 | 71.2 | 90.5 | 67.2 | 73.2 | 91.5 | | | | Res.50 / 23MB |
| PCL(Li et al., 2021) | 39.92 | 56.12 | 68.24 | 87.19 | 55.9 | 60.2 | 77.2 | 71.5 | 76.1 | 93.2 | 59.5 | 66.5 | 86.7 | 62.2 | 70.5 | 90.5 | | | | Res.50 / 47MB |
| SwAV(Caron et al., 2020) | 40.19 | 58.29 | 69.29 | 88.39 | 58.2 | 61.0 | 79.4 | 72.1 | 75.8 | 92.9 | 65.4 | 73.1 | 91.2 | 65.7 | 75.3 | 91.5 | | | | Res.50 / 23MB |
| HCL(Robinson et al., 2021) | 41.39 | 59.92 | 70.29 | 88.39 | 55.9 | 60.8 | 79.3 | 70.2 | 74.6 | 92.3 | 64.2 | 71.2 | 91.2 | 67.2 | 71.7 | 90.7 | | | | Res.50 / 23MB |
| GCA(Chen et al., 2024b) | 44.27 | 60.53 | 74.57 | 91.69 | 60.5 | 63.4 | 79.4 | 72.8 | 75.6 | 93.2 | 67.4 | 74.5 | 91.9 | 67.8 | 76.6 | 92.9 | | | | Res.50 / 23MB |
| INTL(Weng et al., 2024) | 44.96 | 62.49 | 77.05 | 91.92 | 60.1 | 66.5 | 78.1 | 69.5 | 76.3 | 92.8 | 67.2 | 73.5 | 91.7 | 65.8 | 75.2 | 91.7 | | | | Res.50 / 23MB |
| VADSL (ours, cluster-free) | 45.92 | 63.59 | 77.95 | 92.95 | 61.6 | 67.1 | 79.8 | 73.9 | 76.5 | 93.8 | **68.9** | 73.7 | 92.2 | 68.3 | 76.3 | 92.4 | | | | Res.50 / 23MB |
| VADSL (ours, cluster-used) | **45.96** | **64.19** | **77.95** | **93.19** | **62.1** | **67.3** | **80.9** | **74.5** | **77.8** | **94.8** | 68.8 | **74.9** | **92.9** | **69.8** | **77.6** | **93.8** | | | | Res.50 / 23MB |
| BYOL(Grill et al., 2020) | 37.91 | 57.45 | 67.59 | 87.69 | 57.2 | 62.8 | 77.9 | 72.1 | 76.9 | 93.8 | 66.6 | 71.4 | 91.2 | 68.2 | 74.2 | 92.8 | | | | ViT.16 / 48MB |
| SwAV(Caron et al., 2020) | 42.39 | 58.92 | 73.91 | 90.78 | 60.1 | 62.5 | 80.5 | 74.2 | 77.8 | 94.2 | 64.7 | 71.8 | 91.1 | 69.2 | 75.6 | 91.8 | | | | ViT.16 / 48MB |
| DINO(Caron et al., 2021) | — | — | — | — | 61.5 | 65.7 | 81.8 | 78.2 | 79.2 | 95.5 | 72.3 | 76.1 | 92.4 | 76.2 | 78.2 | 94.2 | | | | ViT.16 / 48MB |
| iBOT(Zhou et al., 2022) | — | — | — | — | 61.5 | 68.2 | 82.2 | 77.5 | 78.5 | 95.2 | 71.5 | 75.0 | 91.9 | 75.2 | 76.0 | 92.6 | | | | ViT.16 / 48MB |
| MTE w/ iBOT(Li et al., 2024) | — | — | — | — | 62.3 | 66.7 | 82.5 | 78.5 | 79.5 | 94.8 | 72.4 | 75.4 | 93.3 | **78.3** | **83.9** | 95.2 | | | | ViT.16 / 48MB |
| VADSL (ours, cluster-free) | 46.56 | 65.34 | 78.59 | 93.44 | 63.4 | 68.4 | 82.8 | 79.5 | 81.3 | 96.5 | 72.7 | **77.5** | 93.1 | 79.1 | 82.1 | 95.6 | | | | ViT.16 / 48MB |
| VADSL (ours, cluster-used) | **47.12** | **65.45** | **79.69** | **93.78** | **64.6** | **69.8** | **83.9** | **80.7** | **83.4** | **97.8** | **73.8** | 79.6 | **93.9** | 80.5 | 83.4 | **96.9** | | | | ViT.16 / 48MB |

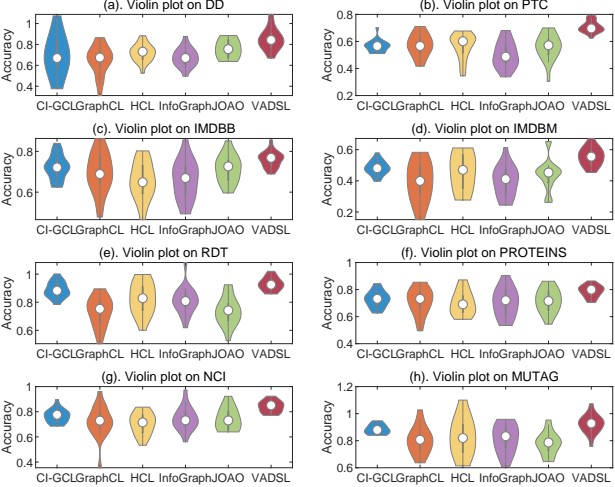

*Figure 6.* Violin plots (with mean values) of compared methods on graph embedding tasks including eight popular datasets.

*Table 5.* Classification accuracy rates (%) of all compared methods on the STS dataset including five tasks and the corresponding average scores.

| METHOD | STS12 | STS13 | STS14 | STS15 | STS16 | Aver. |
|---|---|---|---|---|---|---|
| SimCSE (Gao et al., 2021) | 68.69 | 82.05 | 72.91 | 81.15 | 79.39 | 76.84 |
| PCL (Chen et al., 2022b) | 72.74 | 83.36 | 76.05 | 83.07 | 79.26 | 78.90 |
| Inf.Min (Chen et al., 2022b) | 70.22 | 83.48 | 75.51 | 81.72 | 79.88 | 78.16 |
| miCSE (Klein & Nabi, 2022) | 71.71 | 83.09 | 75.46 | 83.13 | 80.22 | 78.72 |
| SCL (Wu et al., 2022b) | 72.86 | 84.91 | 76.79 | 84.35 | 81.74 | 80.13 |
| ADNCE (Wu et al., 2024) | **72.83** | 81.88 | 74.43 | 85.88 | 81.88 | 79.38 |
| VADSL (ours) | 72.74 | **85.54** | **78.32** | **87.85** | **82.44** | **81.38** |

all eight datasets. Moreover, compared with other graph contrastive learning approaches, JOAO, CI-GCL, and our method can perform relatively better. In most cases, our VADSL surpasses all compared methods with higher accuracy mean and lower accuracy variance.

## 6. Conclusion

In this paper, we introduced an extension to the conventional data point by representing each instance as a data ball, endowing it with a volume value. This led to the natural definition of the VAD metric, which computes the geometric proximity between data balls, allowing the relationship among unsampled instances within these data balls to be effectively captured. The incorporation of a volume expansion regularizer further emphasized the utility of volume-awareness in enhancing the model generalizability. VAD is a general technique that can be easily integrated into both supervised and unsupervised learning tasks with negligible computational overhead. To the best of our knowledge, this is the first work in similarity learning that considers the instance volume. We provided comprehensive theoretical analyses that guarantee the effectiveness of our method. Experiments on real-world data across multiple domains indicated that our learning algorithm acquires more reliable features than state-of-the-art methods. In the future, we plan to apply VAD in broader paradigms of machine learning.

most cases, where our method also achieves the best average score in all compared methods. These results suggest that our method can work well for the text data and VAD leads to enhanced semantic understanding.

**Graph Data.** We further evaluate our method on a challenging graph embedding task using biochemical-molecule data and social-network data, including *DD*, *PTC*, *IMDB-B*, *IMDB-M*, *RDT-B*, *PROTEINS*, *NCI1*, and *MUTAG* (Yanardag & Vishwanathan, 2015). We use the representative method *InfoGraph* (Sun et al., 2020a) as the baseline and perform downstream graph-level classification on these datasets. For evaluation, we fine-tune an *SVM* (Cortes & Vapnik, 1995) on the learned feature representations using 10-fold cross-validation. The dataset is split into training, test, and validation sets in an $8/1/1$ ratio. The accuracy results are reported after 10 runs. Our compared methods include *HCL*, *GraphCL* (You et al., 2020), *JOAO* (You et al., 2021), and *CI-GCL* (Tan et al., 2024). From *violin plots* in Fig. 6, VADSL consistently improves InfoGraph across

## Impact Statement

This paper presents work whose goal is to advance the field of Machine Learning. There are many potential societal consequences of our work, none which we feel must be specifically highlighted here.

## Acknowledgment

The authors would like to thank the anonymous reviewers for their critical and constructive comments and suggestions. This work was supported by the National Science Fund of China under Grant Nos. U24A20330, 62361166670, 62336003, and 12371510.

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

# Appendix for
# "Volume-Aware Distance for Robust Similarity Learning"

## Appendix

This part is the appendix of our manuscript. It includes the additional experiments and the mathematical proofs of theorems.

## A Additional Experiments

### A.1 Additional Experiments on COCO dataset

We would like to further investigate the transferability of our method on the object detection and instance segmentation tasks. We first pre-train the model (with ResNet-50 backbone) on ImageNet-1K, and then fine-tune the pre-trained backbone on the new dataset. Specifically, we select COCO (Lin et al., 2014) as our target dataset and follow the common setting (as discussed in *MoCo-v3* (Chen et al., 2021)) to fine-tune *all layers* of the pre-trained model over the *train2017* set while evaluating the performance on the *val2017* set. We employ *Faster R-CNN* (Ren et al., 2015) and *Mask R-CNN* (He et al., 2017) as our backbone for detection and segmentation, respectively. We implement our method based on the loss functions of SimCLR (negative-used) and BYOL (negative-free). As listed in Tab. A1, our VADSL shows considerable improvement over MoCo-v3 and DINO on both two recognition tasks. This indicates that our method not only works well on classification-oriented tasks but also on more natural image-related recognition tasks.

*Table A1.* Performance of all methods for two transfer learning tasks: object detection and instance segmentation on COCO dataset.

| METHOD | Object Detection | | | Instance Segmentation | | |
|---|---|---|---|---|---|---|
| | $AP^{bb}$ | $AP^{bb}_{50}$ | $AP^{bb}_{75}$ | $AP^{mk}$ | $AP^{mk}_{50}$ | $AP^{mk}_{75}$ |
| Supervised | 38.2 | 59.1 | 41.5 | 35.4 | 56.5 | 38.1 |
| BYOL (Grill et al., 2020) | 39.9 | 60.2 | 43.3 | 36.5 | 58.4 | 39.1 |
| SwAV (Caron et al., 2020) | 40.3 | 61.5 | 44.4 | 36.3 | 58.7 | 39.4 |
| MoCo-v2 (Chen et al., 2020) | 37.6 | 57.9 | 40.8 | 35.3 | 55.9 | 37.9 |
| MoCo-v3 (Chen et al., 2021) | 39.9 | 61.2 | 43.2 | 36.5 | 58.1 | 38.8 |
| DenseCL (Wang et al., 2021) | 40.3 | 59.9 | 44.3 | 36.4 | 57.0 | 39.2 |
| DINO (Caron et al., 2021) | 40.3 | 62.0 | 44.1 | 36.8 | 58.8 | 39.2 |
| INTL (Weng et al., 2024) | 40.7 | 60.9 | 43.7 | 35.4 | 57.3 | 37.6 |
| **VADSL** (neg.-used) | **43.1** | **63.3** | 43.4 | **37.6** | **59.4** | **40.2** |
| **VADSL** (neg.-free) | 42.1 | 63.1 | **45.1** | 36.5 | 58.5 | 39.7 |

### A.2 Additional Experiments on BookCorpus dataset

For the BookCorpus dataset which includes six sub-tasks *movie review sentiment* (MR), *product reviews* (CR), s*ubjectivity classification* (SUBJ), *opinion polarity* (MPQA), *question type classification* (TREC), and *paraphrase identification* (MSRP), we follow the experimental settings in the baseline method *quick-thought* (QT) (Logeswaran & Lee, 2018) to choose the neighboring sentences as positive pairs. Then, we further compare our VDASL with *DCL*, *HCL*, *CO2* (Wei et al., 2021), *UnReMix* (Tabassum et al., 2022), and *ADNCE* (Wu et al., 2024), and the corresponding average classification accuracy rates are shown in Fig. A1.

### A.3 Running Time Comparison

In our learning framework, we have an additional measure-head network as well as the corresponding VER regularizer in the learning objective. We would like to investigate if the efficiency of the learning algorithm will be affected by the additional calculations. Here we further provide experiments to record the training time of our method as well as the corresponding baseline method. Specifically, we use two NVIDIA TeslaV100 GPUs to train our method based on SimCLR and SwAV with 100 epochs, respectively. For each case, we set the batch size to 512 and 1024.

In Tab. A2, we can find that the proposed new measure-head and the new regularizer only brings in very little additional

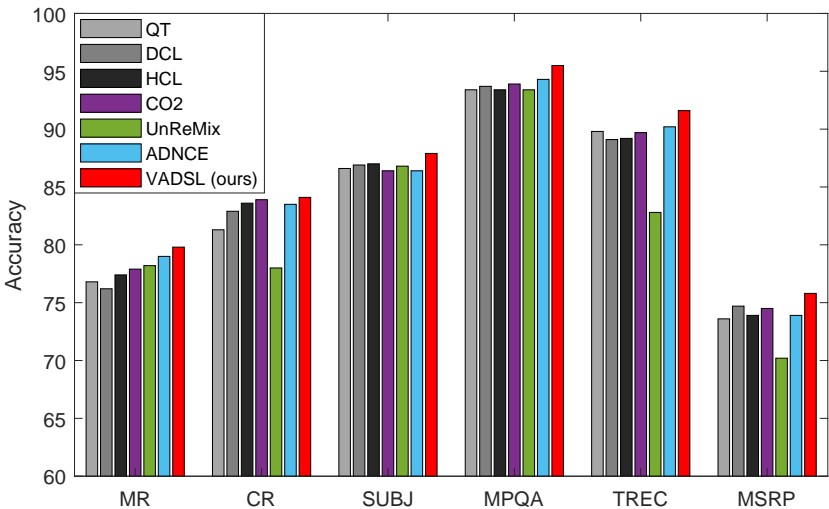

*Figure A1.* Accuracy rates (%) of all methods on BookCorpus dataset including six text classification tasks.

time consumption. This is because the calculations of measure-head and VER are independent to the size of training data, so the training time is completely acceptable in practice use.

### A.4 Parametric Sensitivity

As we only introduced one additional hyper-parameter in our method. Here we want to simply investigate the parametric sensitivity of the regularization parameter $\lambda$ in our learning objective. Specifically, we change $\lambda$ in $[0.01, 5]$, and we record the classification accuracy of our method on STL-10 and CIFAR-10 datasets (batch size=256/512/1024, epochs=100). Tab. A3 shows that the accuracy variation of our method is smaller than $1.5\%$. These results clearly demonstrate that the regularization parameter $\lambda$ is relatively stable within a given range. It implies that the hyper-parameter of our method can be easily tuned in practice use.

*Table A2.* Training time of the baseline methods and our proposed method (100 epochs, in hours).

| METHOD | CIFAR-10 | | ImageNet-100 | | ImageNet-1K | |
|---|---|---|---|---|---|---|
| | 512 | 1024 | 512 | 1024 | 512 | 1024 |
| SimCLR | 2.3 | 1.3 | 10.9 | 5.5 | 70.1 | 35.2 |
| SwAV | 2.6 | 1.7 | 11.5 | 5.8 | 71.2 | 36.7 |
| VADSL (SimCLR+) | 2.4 | 1.5 | 11.2 | 5.6 | 71.5 | 35.6 |
| VADSL (SwAV+) | 2.7 | 1.9 | 11.9 | 6.0 | 72.1 | 36.9 |

*Table A3.* Parametric sensitivity of $\lambda$ on the STL-10 and CIFAR-10 datasets (%). Here $\lambda$ is changed within $[0.01, 5]$.

| dataset (batchsize) | 0.01 | 0.1 | 0.5 | 1.5 | 5 |
|---|---|---|---|---|---|
| STL-10 (256) | 76.8 | 77.8 | **78.1** | 77.9 | 76.9 |
| STL-10 (512) | 78.1 | 78.8 | **79.5** | 78.5 | 78.2 |
| STL-10 (1024) | 81.5 | 81.9 | **82.1** | 81.9 | 81.5 |
| CIFAR-10 (256) | 87.9 | 88.5 | **89.3** | 88.9 | 88.5 |
| CIFAR-10 (512) | 91.5 | 91.9 | 92.3 | **92.5** | 91.8 |
| CIFAR-10 (1024) | 93.2 | 93.6 | **94.5** | 93.6 | 93.1 |

# B Proofs

## B.1 Proof for Theorem 1

*Proof.* For the gradient boundness of $\mathcal{F}$, let us firstly assume that $\max\{\|\nabla\varphi_1(\boldsymbol{x})\|_2, \|\nabla\varphi_2(\boldsymbol{x})\|_2, \ldots, \|\nabla\varphi_m(\boldsymbol{x})\|_2\} \leq \delta$ for any $\boldsymbol{x} \in \mathbb{R}^d$. Then we have that

$$
\begin{aligned}
&\nabla_{\boldsymbol{\varphi}}\mathcal{L}_{\text{emp}}(\boldsymbol{\varphi}, \mathcal{H})\\
&= \mathbb{E}_{\boldsymbol{x}, \{b_j\}_{j=1}^n}\left[-\frac{\sum_{j=0}^n e^{-\mathcal{D}_{\boldsymbol{\varphi}, \mathcal{V}}(\boldsymbol{x}, \boldsymbol{x}_{b_j})/\gamma}}{e^{-\mathcal{D}_{\boldsymbol{\varphi}, \mathcal{V}}(\boldsymbol{x}, \boldsymbol{x}^+)/\gamma}} \cdot \left(\frac{-\frac{1}{\gamma}e^{-\mathcal{D}_{\boldsymbol{\varphi}, \mathcal{V}}(\boldsymbol{x}, \boldsymbol{x}^+)/\gamma}\sum_{j=0}^n e^{-\mathcal{D}_{\boldsymbol{\varphi}, \mathcal{V}}(\boldsymbol{x}, \boldsymbol{x}_{b_j})/\gamma}\nabla_{\boldsymbol{\varphi}}\mathcal{D}_{\boldsymbol{\varphi}, \mathcal{V}}(\boldsymbol{x}, \boldsymbol{x}^+)}{\left(\sum_{j=0}^n e^{-\mathcal{D}_{\boldsymbol{\varphi}, \mathcal{V}}(\boldsymbol{x}, \boldsymbol{x}_{b_j})/\gamma}\right)^2}\right.\right.\\
&\quad\left.\left. - \frac{-\frac{1}{\gamma}e^{-\mathcal{D}_{\boldsymbol{\varphi}, \mathcal{V}}(\boldsymbol{x}, \boldsymbol{x}^+)/\gamma}\sum_{j=0}^n e^{-\mathcal{D}_{\boldsymbol{\varphi}, \mathcal{V}}(\boldsymbol{x}, \boldsymbol{x}_{b_j})/\gamma}\nabla_{\boldsymbol{\varphi}}\mathcal{D}_{\boldsymbol{\varphi}, \mathcal{V}}(\boldsymbol{x}, \boldsymbol{x}_{b_j})}{\left(\sum_{j=0}^n e^{-\mathcal{D}_{\boldsymbol{\varphi}, \mathcal{V}}(\boldsymbol{x}, \boldsymbol{x}_{b_j})/\gamma}\right)^2}\right)\right]\\
&= \mathbb{E}_{\boldsymbol{x}, \{b_j\}_{j=1}^n}\left[\frac{-\frac{1}{\gamma}e^{-\mathcal{D}_{\boldsymbol{\varphi}, \mathcal{V}}(\boldsymbol{x}, \boldsymbol{x}^+)/\gamma}\sum_{j=0}^n e^{-\mathcal{D}_{\boldsymbol{\varphi}, \mathcal{V}}(\boldsymbol{x}, \boldsymbol{x}_{b_j})/\gamma}(\nabla_{\boldsymbol{\varphi}}\mathcal{D}_{\boldsymbol{\varphi}, \mathcal{V}}(\boldsymbol{x}, \boldsymbol{x}^+) - \nabla_{\boldsymbol{\varphi}}\mathcal{D}_{\boldsymbol{\varphi}, \mathcal{V}}(\boldsymbol{x}, \boldsymbol{x}_{b_j}))}{e^{-\mathcal{D}_{\boldsymbol{\varphi}, \mathcal{V}}(\boldsymbol{x}, \boldsymbol{x}^+)/\gamma}\left(\sum_{j=0}^n e^{-\mathcal{D}_{\boldsymbol{\varphi}, \mathcal{V}}(\boldsymbol{x}, \boldsymbol{x}_{b_j})/\gamma}\right)}\right],
\end{aligned}
$$

and thus[1]

$$
\begin{aligned}
&\|\nabla_{\boldsymbol{\varphi}}\mathcal{L}_{\text{emp}}(\boldsymbol{\varphi}, \mathcal{H})\|_2^2\\
&\leq \max\left(\left|\frac{-\frac{1}{\gamma}e^{-\mathcal{D}_{\boldsymbol{\varphi}, \mathcal{V}}(\boldsymbol{x}, \boldsymbol{x}^+)/\gamma}\sum_{j=0}^n e^{-\mathcal{D}_{\boldsymbol{\varphi}, \mathcal{V}}(\boldsymbol{x}, \boldsymbol{x}_{b_j})/\gamma}}{e^{-\mathcal{D}_{\boldsymbol{\varphi}, \mathcal{V}}(\boldsymbol{x}, \boldsymbol{x}^+)/\gamma}\left(\sum_{j=0}^n e^{-\mathcal{D}_{\boldsymbol{\varphi}, \mathcal{V}}(\boldsymbol{x}, \boldsymbol{x}_{b_j})/\gamma}\right)}\right|^2\right)\left\|n\nabla_{\boldsymbol{\varphi}}\mathcal{D}_{\boldsymbol{\varphi}, \mathcal{V}}(\boldsymbol{x}, \boldsymbol{x}^+) - \sum_{j=0}^n \nabla_{\boldsymbol{\varphi}}\mathcal{D}_{\boldsymbol{\varphi}, \mathcal{V}}(\boldsymbol{x}, \boldsymbol{x}_{b_j})\right\|_2^2\\
&\leq \frac{n^2}{\gamma^2}(n+2)^2\max\left(\|\nabla_{\boldsymbol{\varphi}}\mathcal{D}_{\boldsymbol{\varphi}, \mathcal{V}}(\boldsymbol{x}, \boldsymbol{x}^+)\|_2^2, \|\nabla_{\boldsymbol{\varphi}}\mathcal{D}_{\boldsymbol{\varphi}, \mathcal{V}}(\boldsymbol{x}, \boldsymbol{x}_{b_1})\|_2^2, \ldots, \|\nabla_{\boldsymbol{\varphi}}\mathcal{D}_{\boldsymbol{\varphi}, \mathcal{V}}(\boldsymbol{x}, \boldsymbol{x}_{b_n})\|_2^2\right)\\
&\leq \frac{n^2}{\gamma^2}(n+2)^2\delta_D\\
&= \delta_1.
\end{aligned}
$$

Meanwhile we have that

$$
\begin{aligned}
&\nabla_{\boldsymbol{\varphi}}\mathcal{R}_{\text{expand}}(\boldsymbol{\varphi}, \mathcal{H})\\
&= \sum_{j=1}^n -e^{-\mathcal{V}(\boldsymbol{x}_{b_j})}\frac{\partial\mathcal{V}(\boldsymbol{x}_{b_j})}{\partial\mathcal{H}}\nabla\boldsymbol{\varphi}(\boldsymbol{x}),
\end{aligned}
\tag{0.1}
$$

and thus

$$
\begin{aligned}
&\|\nabla_{\boldsymbol{\varphi}}\mathcal{R}_{\text{expand}}(\boldsymbol{\varphi}, \mathcal{H})\|_2^2\\
&\leq \left\|\sum_{j=1}^n \frac{\partial\mathcal{V}(\boldsymbol{x}_{b_j})}{\partial\mathcal{H}}\nabla\boldsymbol{\varphi}(\boldsymbol{x})\right\|_2^2\\
&\leq \sum_{j=1}^n \left\|\frac{\partial\mathcal{V}(\boldsymbol{x}_{b_j})}{\partial\mathcal{H}}\nabla\boldsymbol{\varphi}(\boldsymbol{x})\right\|_2^2\\
&\leq \sum_{j=1}^n \|\nabla\boldsymbol{\varphi}(\boldsymbol{x})\|_2^2\\
&= n\delta^2\\
&= \delta_2.
\end{aligned}
\tag{0.2}
$$

---

[1] For details, here $\|\nabla_{\boldsymbol{\varphi}}\mathcal{D}_{\boldsymbol{\varphi}, \mathcal{V}}(\boldsymbol{x}, \widehat{\boldsymbol{x}})\|_2^2 \leq \max_{(\boldsymbol{x}, \widehat{\boldsymbol{x}})}\left(\frac{\nabla_{\boldsymbol{\varphi}}\|\boldsymbol{\varphi}(\boldsymbol{x})-\boldsymbol{\varphi}(\widehat{\boldsymbol{x}})\|_2^2(e^{\mathcal{V}(\boldsymbol{x})+\mathcal{V}(\widehat{\boldsymbol{x}})})^2 - 2\|\boldsymbol{\varphi}(\boldsymbol{x})-\boldsymbol{\varphi}(\widehat{\boldsymbol{x}})\|_2^2(e^{\mathcal{V}(\boldsymbol{x})+\mathcal{V}(\widehat{\boldsymbol{x}})})(\nabla\mathcal{H}\nabla\boldsymbol{\varphi}(\boldsymbol{x})+\nabla\mathcal{H}\nabla\boldsymbol{\varphi}(\widehat{\boldsymbol{x}}))}{(e^{\mathcal{V}(\boldsymbol{x})+\mathcal{V}(\widehat{\boldsymbol{x}})})^4}\right) \leq$
$\max(\|\boldsymbol{\varphi}(\widehat{\boldsymbol{x}})\sum_{j=1}^n \nabla\varphi_j(\boldsymbol{x}) + \boldsymbol{\varphi}(\boldsymbol{x})\sum_{j=1}^n \nabla\varphi_j(\widehat{\boldsymbol{x}})\|_2^2) + 2\max(\|\boldsymbol{\varphi}(\boldsymbol{x})-\boldsymbol{\varphi}(\widehat{\boldsymbol{x}})\|_2^2\|\nabla\boldsymbol{\varphi}(\boldsymbol{x})+\nabla\boldsymbol{\varphi}(\widehat{\boldsymbol{x}})\|_2^2) \leq 4n^2\delta^2 + 8\cdot 2\delta^2 = \delta_D.$

Similarly, we can obtain that

$$\|\nabla_{\mathcal{H}}\mathcal{L}_{\text{emp}}(\boldsymbol{\varphi}, \mathcal{H})\|_2^2 \leq \delta_3, \quad \text{and} \quad \|\nabla_{\mathcal{H}}\mathcal{R}_{\text{emp}}(\boldsymbol{\varphi}, \mathcal{H})\|_2^2 \leq \delta_4. \tag{0.3}$$

Finally, we have that

$$\begin{aligned}
&\|\nabla\mathcal{F}(\boldsymbol{\varphi}, \mathcal{H})\|_2^2 \\
&= \|\nabla\mathcal{L}_{\text{emp}}(\boldsymbol{\varphi}, \mathcal{H}) + \lambda\nabla\mathcal{R}_{\text{expand}}(\boldsymbol{\varphi}, \mathcal{H})\|_2^2 \\
&= \|(\nabla_{\boldsymbol{\varphi}}\mathcal{L}_{\text{emp}}(\boldsymbol{\varphi}, \mathcal{H}) + \lambda\nabla_{\boldsymbol{\varphi}}\mathcal{R}_{\text{expand}}(\boldsymbol{\varphi}, \mathcal{H}), \nabla_{\mathcal{H}}\mathcal{L}_{\text{emp}}(\boldsymbol{\varphi}, \mathcal{H}) + \lambda\nabla_{\mathcal{H}}\mathcal{R}_{\text{expand}}(\boldsymbol{\varphi}, \mathcal{H}))\|_2^2 \\
&\leq \|\nabla_{\boldsymbol{\varphi}}\mathcal{L}_{\text{emp}}(\boldsymbol{\varphi}, \mathcal{H})\|_2^2 + \lambda^2\|\nabla_{\boldsymbol{\varphi}}\mathcal{R}_{\text{expand}}(\boldsymbol{\varphi}, \mathcal{H})\|_2^2 + \|\nabla_{\mathcal{H}}\mathcal{L}_{\text{emp}}(\boldsymbol{\varphi}, \mathcal{H})\|_2^2 + \lambda^2\|\nabla_{\mathcal{H}}\mathcal{R}_{\text{expand}}(\boldsymbol{\varphi}, \mathcal{H})\|_2^2 \\
&= \delta_1 + \lambda^2\delta_2 + \delta_3 + \lambda^2\delta_4, \tag{0.4}
\end{aligned}$$

which shows that $\mathcal{F}(\boldsymbol{\varphi}, \mathcal{H})$ is always gradient-bounded.

For the Lipschitz-smoothness of $\mathcal{F}$, let us also assume that there exists $L > 0$ such that $\|\nabla\varphi(\boldsymbol{x}) - \nabla\widehat{\varphi}(\boldsymbol{x})\|_2 \leq L\|\varphi - \widehat{\varphi}\|_2$. Then we have that

$$\begin{aligned}
&\left\|\nabla_{\boldsymbol{\varphi}, \mathcal{H}}\mathcal{L}_{\text{emp}}(\boldsymbol{\varphi}, \mathcal{H}) - \nabla_{\boldsymbol{\varphi}, \mathcal{H}}\mathcal{L}_{\text{emp}}(\widehat{\boldsymbol{\varphi}}, \widehat{\mathcal{H}})\right\|_2^2 \\
&= \left\|\nabla_{\boldsymbol{\varphi}}\mathcal{L}_{\text{emp}}(\boldsymbol{\varphi}, \mathcal{H}) - \nabla_{\boldsymbol{\varphi}}\mathcal{L}_{\text{emp}}(\widehat{\boldsymbol{\varphi}}, \widehat{\mathcal{H}})\right\|_2^2 + \left\|\nabla_{\mathcal{H}}\mathcal{L}_{\text{emp}}(\boldsymbol{\varphi}, \mathcal{H}) - \nabla_{\mathcal{H}}\mathcal{L}_{\text{emp}}(\widehat{\boldsymbol{\varphi}}, \widehat{\mathcal{H}})\right\|_2^2 \\
&\leq \left\|\max(\|\boldsymbol{\varphi}(\widehat{\boldsymbol{x}})\sum_{j=1}^n \nabla\varphi_j(\boldsymbol{x}) + \boldsymbol{\varphi}(\boldsymbol{x})\sum_{j=1}^n \nabla\varphi_j(\widehat{\boldsymbol{x}})\|_2^2) + 2\max(\|\boldsymbol{\varphi}(\boldsymbol{x}) - \boldsymbol{\varphi}(\widehat{\boldsymbol{x}})\|_2^2 \|\nabla\boldsymbol{\varphi}(\boldsymbol{x}) + \nabla\boldsymbol{\varphi}(\widehat{\boldsymbol{x}})\|_2^2)\right. \\
&\left. \quad - \max(\|\widehat{\boldsymbol{\varphi}}(\widehat{\boldsymbol{x}})\sum_{j=1}^n \nabla\widehat{\varphi}_j(\boldsymbol{x}) + \widehat{\boldsymbol{\varphi}}(\boldsymbol{x})\sum_{j=1}^n \nabla\widehat{\varphi}_j(\widehat{\boldsymbol{x}})\|_2^2) - 2\max(\|\widehat{\boldsymbol{\varphi}}(\boldsymbol{x}) - \widehat{\boldsymbol{\varphi}}(\widehat{\boldsymbol{x}})\|_2^2 \|\nabla\widehat{\boldsymbol{\varphi}}(\boldsymbol{x}) + \nabla\widehat{\boldsymbol{\varphi}}(\widehat{\boldsymbol{x}})\|_2^2)\right\|_2^2 \\
&\quad + 1 \cdot \|\mathcal{H} - \widehat{\mathcal{H}}\|_2^2 \\
&\leq \max(2\|\boldsymbol{\varphi}(\widehat{\boldsymbol{x}}) - \widehat{\boldsymbol{\varphi}}(\widehat{\boldsymbol{x}})\|_2^2 \sum_{j=1}^n \|\nabla\varphi_j(\boldsymbol{x}) - \nabla\widehat{\varphi}_j(\boldsymbol{x})\|_2^2) + 2\max(\|\nabla\boldsymbol{\varphi}(\boldsymbol{x}) - \nabla\widehat{\boldsymbol{\varphi}}(\boldsymbol{x})\|_2 + \|\nabla\boldsymbol{\varphi}(\widehat{\boldsymbol{x}}) - \nabla\widehat{\boldsymbol{\varphi}}(\widehat{\boldsymbol{x}})\|_2) \\
&\quad + 1 \cdot \|\mathcal{H} - \widehat{\mathcal{H}}\|_2^2 \\
&\leq 4nL\|\varphi - \widehat{\varphi}\|_2 + 4L\|\varphi - \widehat{\varphi}\|_2 + 1 \cdot \|\mathcal{H} - \widehat{\mathcal{H}}\|_2^2 \\
&\leq 4(n+1)L\|(\varphi, \mathcal{H}) - (\widehat{\varphi}, \widehat{\mathcal{H}})\|_2^2, \tag{0.5}
\end{aligned}$$

and the proof is completed. $\qquad\qquad\qquad\qquad\qquad\qquad\qquad\qquad\qquad\qquad\qquad\qquad\qquad\qquad\qquad\quad \square$

## B.2 Proof for Theorem 2

We first introduce the following lemma for proving Theorem 2.

**Lemma 1.** *For independent random variables $t_1, t_2, \ldots, t_n \in \mathcal{T}$ and a given function $\omega : \mathcal{T}^n \to \mathbb{R}$, if $\forall v_i' \in \mathcal{T}$ $(i = 1, 2, \ldots, n)$, the function satisfies*

$$|\omega(t_1, \ldots, t_i, \ldots, t_n) - \omega(t_1, \ldots, t_i', \ldots, t_n)| \leq \rho_i, \tag{0.6}$$

*then for any given $\mu > 0$, it holds that $P\{|\omega(t_1, \ldots, t_n) - \mathbb{E}[\omega(t_1, \ldots, t_n)]| > \mu\} \leq 2e^{-2\mu^2/\sum_{i=1}^n \rho_i^2}$.*

*Proof.* We prove Theorem 2 by analyzing the perturbation (i.e., $\rho_i$ in the above Eq. (0.6)) of the loss function $\mathcal{L}_{\text{emp}}$.

We denote that

$$\omega = \mathcal{L}_{\text{emp}}(\boldsymbol{\varphi}, \mathcal{H}; \mathcal{X}) = \frac{1}{N}\sum_{i=1}^N -\log\frac{e^{-\mathcal{D}_{\boldsymbol{\varphi}, \mathcal{H}}(\boldsymbol{x}_i, \boldsymbol{x}^+)/\gamma}}{e^{-\mathcal{D}_{\boldsymbol{\varphi}, \mathcal{H}}(\boldsymbol{x}_i, \boldsymbol{x}^+)/\gamma} + \sum_{j=1}^n e^{-\mathcal{D}_{\boldsymbol{\varphi}, \mathcal{H}}(\boldsymbol{x}_i, \boldsymbol{x}_{b_j})/\gamma}}, \tag{0.7}$$

and

$$\widetilde{\omega_r} = \frac{1}{N}\left[\left(\sum_{i\neq r}^{N} -\log\frac{e^{-\mathcal{D}_{\varphi,\mathcal{H}}(\boldsymbol{x}_i,\boldsymbol{x}^+)/\gamma}}{e^{-\mathcal{D}_{\varphi,\mathcal{H}}(\boldsymbol{x}_i,\boldsymbol{x}^+)/\gamma}+\sum_{j=1}^{n}e^{-\mathcal{D}_{\varphi,\mathcal{H}}(\boldsymbol{x}_i,\boldsymbol{x}_{b_j})/\gamma}},\right) - \log\frac{e^{-\mathcal{D}_{\varphi,\mathcal{H}}(\widehat{\boldsymbol{x}},\widehat{\boldsymbol{x}}^+)/\gamma}}{e^{-\mathcal{D}_{\varphi,\mathcal{H}}(\widehat{\boldsymbol{x}},\widehat{\boldsymbol{x}}^+)/\gamma}+\sum_{j=1}^{n}e^{-\mathcal{D}_{\varphi,\mathcal{H}}(\widehat{\boldsymbol{x}},\widehat{\boldsymbol{x}}_{b_j})/\gamma}}\right],$$

(0.8)

where $(\widehat{\boldsymbol{x}}, \{\widehat{\boldsymbol{x}}_{b_j}\}_{j=1}^{n})$ is an arbitrary mini-batch from the sample space. Then we have that

$$|\omega - \widetilde{\omega_r}|$$
$$= \frac{1}{N}\left|\log\frac{e^{-\mathcal{D}_{\varphi,\mathcal{H}}(\widehat{\boldsymbol{x}},\widehat{\boldsymbol{x}}^+)/\gamma}}{e^{-\mathcal{D}_{\varphi,\mathcal{H}}(\widehat{\boldsymbol{x}},\widehat{\boldsymbol{x}}^+)/\gamma}+\sum_{j=1}^{n}e^{-\mathcal{D}_{\varphi,\mathcal{H}}(\widehat{\boldsymbol{x}},\widehat{\boldsymbol{x}}_{b_j})/\gamma}} - \log\frac{e^{-\mathcal{D}_{\varphi,\mathcal{H}}(\boldsymbol{x}_r,\boldsymbol{x}^+)/\gamma}}{e^{-\mathcal{D}_{\varphi,\mathcal{H}}(\boldsymbol{x}_r,\boldsymbol{x}^+)/\gamma}+\sum_{j=1}^{n}e^{-\mathcal{D}_{\varphi,\mathcal{H}}(\boldsymbol{x}_r,\boldsymbol{x}_{b_j})/\gamma}}\right|$$
$$\leq \frac{1}{N}\log\left[\frac{e^{-\mathcal{D}_{\varphi,\mathcal{H}}(\widehat{\boldsymbol{x}},\widehat{\boldsymbol{x}}^+)/\gamma}(e^{-\mathcal{D}_{\varphi,\mathcal{H}}(\boldsymbol{x}_r,\boldsymbol{x}^+)/\gamma}+\sum_{j=1}^{n}e^{-\mathcal{D}_{\varphi,\mathcal{H}}(\boldsymbol{x}_r,\boldsymbol{x}_{b_j})/\gamma})}{e^{-\mathcal{D}_{\varphi,\mathcal{H}}(\boldsymbol{x}_r,\boldsymbol{x}^+)/\gamma}(e^{-\mathcal{D}_{\varphi,\mathcal{H}}(\widehat{\boldsymbol{x}},\widehat{\boldsymbol{x}}^+)/\gamma}+\sum_{j=1}^{n}e^{-\mathcal{D}_{\varphi,\mathcal{H}}(\widehat{\boldsymbol{x}},\widehat{\boldsymbol{x}}_{b_j})/\gamma})}\right]$$
$$\leq \frac{\omega(n)\log(1+\max\{d_{\varphi}(\boldsymbol{t},\widehat{\boldsymbol{t}})|\boldsymbol{t},\widehat{\boldsymbol{t}}\in\mathscr{X}\})}{2C\lambda N},$$

(0.9)

where $\omega(n) = \log\left(\frac{e^2}{n}+1\right)$. Meanwhile, we have

$$\frac{1}{N}\sum_{i=1}^{N} -\log\frac{e^{-\mathcal{D}_{\varphi,\mathcal{H}}(\boldsymbol{x}_i,\boldsymbol{x}^+)/\gamma}}{e^{-\mathcal{D}_{\varphi,\mathcal{H}}(\boldsymbol{x}_i,\boldsymbol{x}^+)/\gamma}+\sum_{j=1}^{n}e^{-\mathcal{D}_{\varphi,\mathcal{H}}(\boldsymbol{x}_i,\boldsymbol{x}_{b_j})/\gamma}} - \mathbb{E}\left(-\log\frac{e^{-\mathcal{D}_{\varphi,\mathcal{H}}(\boldsymbol{x}_i,\boldsymbol{x}^+)/\gamma}}{e^{-\mathcal{D}_{\varphi,\mathcal{H}}(\boldsymbol{x}_i,\boldsymbol{x}^+)/\gamma}+\sum_{j=1}^{n}e^{-\mathcal{D}_{\varphi,\mathcal{H}}(\boldsymbol{x}_i,\boldsymbol{x}_{b_j})/\gamma}}\right)$$
$$= \mathcal{L}_{\text{emp}}(\varphi,\mathcal{H};\mathscr{X}) - \widetilde{\mathcal{L}}_{\text{emp}}(\varphi,\mathcal{H};\mathscr{D}).$$

(0.10)

By Lemma 1, we let that for all $i = 1, 2, \ldots, N$

$$\rho_i = \frac{\omega(n)\log(1+\max\{d_{\varphi}(\boldsymbol{t},\widehat{\boldsymbol{t}})|\boldsymbol{t},\widehat{\boldsymbol{t}}\in\mathscr{X}\})}{2C\lambda N},$$

(0.11)

so that we have

$$P\left\{\left|\mathcal{L}_{\text{emp}}(\varphi;\mathscr{X}) - \widetilde{\mathcal{L}}_{\text{emp}}(\varphi;\mathscr{D})\right| < \frac{\omega(n)\log(1+\max\{d_{\varphi}(\boldsymbol{t},\widehat{\boldsymbol{t}})|\boldsymbol{t},\widehat{\boldsymbol{t}}\in\mathscr{X}\})}{2C\lambda}\sqrt{\frac{\ln(2/\delta)}{2N}}\right\}$$
$$= 1 - 2e^{-2\mu^2/\sum_{i=1}^{N}\rho_i^2}$$
$$\geq 1 - 2e^{\frac{-2N(\eta\sqrt{[\ln(2/\delta)]/(2C\lambda N)})^2}{\max^2(\omega(n)\log(1+\max\{d_{\varphi}(\boldsymbol{t},\widehat{\boldsymbol{t}})|\boldsymbol{t}\in\mathscr{X}\}\alpha)}}$$
$$= 1 - 2e^{-2N\left(\sqrt{[\ln(2/\delta)]/(2C\lambda N)}\right)^2}$$
$$= 1 - 2e^{-\ln(2/\delta)}$$
$$= 1 - \delta,$$

(0.12)

where $\eta = \frac{\omega(n)\log(1+\max\{d_{\varphi}(\boldsymbol{t},\widehat{\boldsymbol{t}})|\boldsymbol{t},\widehat{\boldsymbol{t}}\in\mathscr{X}\})}{2C\lambda}$ and $\mu = \sqrt{[\ln(2/\delta)]/(2\theta(\lambda)N)}$. The proof is completed. $\square$

### B.3 Proof for Theorem 3

*Proof.* We prove the theorem via *mathematical induction*.

**i).** We first validate that the inequality holds for $(a_1, b_1) \succcurlyeq (a_2, b_2)$. To be specific, we suppose that $\mathcal{D}_{\varphi,\mathcal{V}}(\boldsymbol{x}_{a_1},\boldsymbol{x}_{b_1}) < \mathcal{D}_{\varphi,\mathcal{V}}(\boldsymbol{x}_{a_2},\boldsymbol{x}_{b_2})$. Then we let

$$\begin{cases} \mathcal{V}_S(\boldsymbol{x}_{a_2}) = S\mathcal{V}(\boldsymbol{x}_{a_2}), \\ \mathcal{V}_S(\boldsymbol{x}_{b_2}) = S\mathcal{V}(\boldsymbol{x}_{a_2}), \\ \mathcal{V}_S(\boldsymbol{x}_{a_1}) = \mathcal{V}(\boldsymbol{x}_{a_1}), \\ \mathcal{V}_S(\boldsymbol{x}_{b_1}) = \mathcal{V}(\boldsymbol{x}_{b_1}), \end{cases}$$

(0.13)

and we have that

$$\lim_{S \to +\infty} \frac{\mathcal{D}_{\boldsymbol{\varphi}, \mathcal{V}_S}(\boldsymbol{x}_{a_2}, \boldsymbol{x}_{b_2})}{\mathcal{D}_{\boldsymbol{\varphi}, \mathcal{V}_S}(\boldsymbol{x}_{a_1}, \boldsymbol{x}_{b_1})} = \lim_{S \to +\infty} \frac{\|\boldsymbol{\varphi}(\boldsymbol{x}_{a_2}) - \boldsymbol{\varphi}(\boldsymbol{x}_{b_2})\|_2}{\|\boldsymbol{\varphi}(\boldsymbol{x}_{a_1}) - \boldsymbol{\varphi}(\boldsymbol{x}_{b_1})\|_2} \cdot \frac{\mathrm{e}^{\mathcal{V}(\boldsymbol{x}_{a_1}) + \mathcal{V}(\boldsymbol{x}_{b_1})}}{\mathrm{e}^{S(\mathcal{V}(\boldsymbol{x}_{a_2}) + \mathcal{V}(\boldsymbol{x}_{b_2}))}} = 0, \tag{0.14}$$

which implies that there exists sufficiently large $\widehat{S}$ such that

$$\mathcal{D}_{\boldsymbol{\varphi}, \mathcal{V}_{\widehat{S}}}(\boldsymbol{x}_{a_2}, \boldsymbol{x}_{b_2}) \leq \mathcal{D}_{\boldsymbol{\varphi}, \mathcal{V}_{\widehat{S}}}(\boldsymbol{x}_{a_1}, \boldsymbol{x}_{b_1}). \tag{0.15}$$

Therefore, the case of $(a_1, b_1) \succcurlyeq (a_2, b_2)$ clearly holds.

**ii).** By assume that the inequality holds for $(a_1, b_1) \succcurlyeq (a_2, b_2) \succcurlyeq \ldots \succcurlyeq (a_K, b_K)$, we show that the inequality can be still satisfied for $(a_1, b_1) \succcurlyeq (a_2, b_2) \succcurlyeq \ldots \succcurlyeq (a_{K+1}, b_{K+1})$. Specifically, without loss of generality, we assume that

$$\mathcal{D}_{\boldsymbol{\varphi}, \mathcal{V}}(\boldsymbol{x}_{a_1}, \boldsymbol{x}_{b_1}) \geq \mathcal{D}_{\boldsymbol{\varphi}, \mathcal{V}}(\boldsymbol{x}_{a_2}, \boldsymbol{x}_{b_2}) \geq \cdots \geq \mathcal{D}_{\boldsymbol{\varphi}, \mathcal{V}}(\boldsymbol{x}_{a_i}, \boldsymbol{x}_{b_i})$$
$$\geq \mathcal{D}_{\boldsymbol{\varphi}, \mathcal{V}}(\boldsymbol{x}_{a_{K+1}}, \boldsymbol{x}_{b_{K+1}}) \geq \mathcal{D}_{\boldsymbol{\varphi}, \mathcal{V}}(\boldsymbol{x}_{a_{i+1}}, \boldsymbol{x}_{b_{i+1}}) \geq \cdots \geq \mathcal{D}_{\boldsymbol{\varphi}, \mathcal{V}}(\boldsymbol{x}_{a_K}, \boldsymbol{x}_{b_K}), \tag{0.16}$$

where $0 \leq i \leq K - 1$. Then we let

$$\begin{cases} \mathcal{V}_S(\boldsymbol{x}_{a_{K+1}}) = S\mathcal{V}(\boldsymbol{x}_{a_{K+1}}), \\ \mathcal{V}_S(\boldsymbol{x}_{b_{K+1}}) = S\mathcal{V}(\boldsymbol{x}_{b_{K+1}}), \\ \mathcal{V}_S(\boldsymbol{x}_{a_j}) = \mathcal{V}(\boldsymbol{x}_{a_j}), \\ \mathcal{V}_S(\boldsymbol{x}_{b_j}) = \mathcal{V}(\boldsymbol{x}_{b_j}), \end{cases} \tag{0.17}$$

and thus we have

$$\lim_{S \to +\infty} \frac{\mathcal{D}_{\boldsymbol{\varphi}, \mathcal{V}_S}(\boldsymbol{x}_{a_{K+1}}, \boldsymbol{x}_{b_{K+1}})}{\mathcal{D}_{\boldsymbol{\varphi}, \mathcal{V}_S}(\boldsymbol{x}_{a_j}, \boldsymbol{x}_{b_j})}$$
$$= \lim_{S \to +\infty} \frac{\|\boldsymbol{\varphi}(\boldsymbol{x}_{a_{K+1}}) - \boldsymbol{\varphi}(\boldsymbol{x}_{b_{K+1}})\|_2}{\|\boldsymbol{\varphi}(\boldsymbol{x}_{a_j}) - \boldsymbol{\varphi}(\boldsymbol{x}_{b_j})\|_2} \cdot \frac{\mathrm{e}^{\mathcal{V}(\boldsymbol{x}_{a_j})(1 + (S-1)\mathrm{sign}(a_j \in \{a_{K+1}, b_{K+1}\})) + \mathcal{V}(\boldsymbol{x}_{b_j})(1 + (S-1)\mathrm{sign}(b_j \in \{a_{K+1}, b_{K+1}\}))}}{\mathrm{e}^{S(\mathcal{V}(\boldsymbol{x}_{a_{K+1}}) + \mathcal{V}(\boldsymbol{x}_{b_{K+1}}))}}$$
$$= 0, \tag{0.18}$$

where $j = 1, 2, \ldots, K$, and at most one of the conditions $a_j \in \{a_{K+1}, b_{K+1}\}$ and $b_j \in \{a_{K+1}, b_{K+1}\}$ can be satisfied due to the fact that $(a_{K+1}, b_{K+1}) \notin \{(a_1, b_1), (a_2, b_2), \ldots, (a_K, b_K)\}$. Therefore, there exists sufficiently large $S^*$ such that

$$\mathcal{D}_{\boldsymbol{\varphi}, \mathcal{V}_{S^*}}(\boldsymbol{x}_{a_{K+1}}, \boldsymbol{x}_{b_{K+1}}) \leq \min\{\mathcal{D}_{\boldsymbol{\varphi}, \mathcal{V}_{S^*}}(\boldsymbol{x}_{a_1}, \boldsymbol{x}_{b_1}), \mathcal{D}_{\boldsymbol{\varphi}, \mathcal{V}_{S^*}}(\boldsymbol{x}_{a_2}, \boldsymbol{x}_{b_2}), \ldots, \mathcal{D}_{\boldsymbol{\varphi}, \mathcal{V}_{S^*}}(\boldsymbol{x}_{a_K}, \boldsymbol{x}_{b_K})\}. \tag{0.19}$$

Then we need to resort $\{\mathcal{D}_{\boldsymbol{\varphi}, \mathcal{V}_{S^*}}(\boldsymbol{x}_{a_1}, \boldsymbol{x}_{b_1}), \mathcal{D}_{\boldsymbol{\varphi}, \mathcal{V}_{S^*}}(\boldsymbol{x}_{a_2}, \boldsymbol{x}_{b_2}), \ldots, \mathcal{D}_{\boldsymbol{\varphi}, \mathcal{V}_{S^*}}(\boldsymbol{x}_{a_K}, \boldsymbol{x}_{b_K})\}$. To be specific, we further construct that

$$\begin{cases} \mathcal{V}_S^{\sqrt{S}}(\boldsymbol{x}_{a_K}) = \sqrt{S}\mathcal{V}(\boldsymbol{x}_{a_K}), \\ \mathcal{V}_S^{\sqrt{S}}(\boldsymbol{x}_{b_K}) = \sqrt{S}\mathcal{V}(\boldsymbol{x}_{b_K}), \\ \mathcal{V}_S^{\sqrt{S}}(\boldsymbol{x}_{a_j}) = \mathcal{V}_S(\boldsymbol{x}_{a_j}), \\ \mathcal{V}_S^{\sqrt{S}}(\boldsymbol{x}_{b_j}) = \mathcal{V}_S(\boldsymbol{x}_{b_j}), \end{cases} \tag{0.20}$$

and we have that

$$\lim_{S \to +\infty} \frac{\mathcal{D}_{\boldsymbol{\varphi}, \mathcal{V}_S^{\sqrt{S}}}(\boldsymbol{x}_{a_K}, \boldsymbol{x}_{b_K})}{\mathcal{D}_{\boldsymbol{\varphi}, \mathcal{V}_S^{\sqrt{S}}}(\boldsymbol{x}_{a_j}, \boldsymbol{x}_{b_j})}$$
$$= \lim_{S \to +\infty} \frac{\|\boldsymbol{\varphi}(\boldsymbol{x}_{a_K}) - \boldsymbol{\varphi}(\boldsymbol{x}_{b_K})\|_2}{\|\boldsymbol{\varphi}(\boldsymbol{x}_{a_j}) - \boldsymbol{\varphi}(\boldsymbol{x}_{b_j})\|_2} \cdot \frac{\mathrm{e}^{\mathcal{V}(\boldsymbol{x}_{a_j})(1 + (\sqrt{S}-1)\mathrm{sign}(a_j \in \{a_K, b_K\})) + \mathcal{V}(\boldsymbol{x}_{b_j})(1 + (\sqrt{S}-1)\mathrm{sign}(b_j \in \{a_K, b_K\}))}}{\mathrm{e}^{\sqrt{S}(\mathcal{V}(\boldsymbol{x}_{a_K}) + \mathcal{V}(\boldsymbol{x}_{b_K}))}}$$
$$= 0, \tag{0.21}$$

where $j = 1, 2, \ldots, K - 1$, and at most one of the conditions $a_j \in \{a_K, b_K\}$ and $b_j \in \{a_K, b_K\}$ can be satisfied due to the fact that $(a_K, b_K) \notin \{(a_1, b_1), (a_2, b_2), \ldots, (a_{K-1}, b_{K-1})\}$. Therefore, there exists sufficiently large $S_1$ such that

$$\mathcal{D}_{\boldsymbol{\varphi}, \mathcal{V}_{S_1}^{\sqrt{S_1}}}(\boldsymbol{x}_{a_K}, \boldsymbol{x}_{b_K}) \leq \min\{\mathcal{D}_{\boldsymbol{\varphi}, \mathcal{V}_{S_1}^{\sqrt{S_1}}}(\boldsymbol{x}_{a_1}, \boldsymbol{x}_{b_1}), \mathcal{D}_{\boldsymbol{\varphi}, \mathcal{V}_{S_1}^{\sqrt{S_1}}}(\boldsymbol{x}_{a_2}, \boldsymbol{x}_{b_2}), \ldots, \mathcal{D}_{\boldsymbol{\varphi}, \mathcal{V}_{S_1}^{\sqrt{S_1}}}(\boldsymbol{x}_{a_{K-1}}, \boldsymbol{x}_{b_{K-1}})\}. \tag{0.22}$$

By letting $S^* = S_1$, we have that

$$
\begin{aligned}
&\mathcal{D}_{\boldsymbol{\varphi}, \mathcal{V}_{S^*}^{\sqrt{S^*}}}(\boldsymbol{x}_{a_{K+1}}, \boldsymbol{x}_{b_{K+1}}) \\
&\leq \mathcal{D}_{\boldsymbol{\varphi}, \mathcal{V}_{S^*}^{\sqrt{S^*}}}(\boldsymbol{x}_{a_K}, \boldsymbol{x}_{b_K}) \\
&\leq \min\{\mathcal{D}_{\boldsymbol{\varphi}, \mathcal{V}_{S^*}^{\sqrt{S^*}}}(\boldsymbol{x}_{a_1}, \boldsymbol{x}_{b_1}), \mathcal{D}_{\boldsymbol{\varphi}, \mathcal{V}_{S^*}^{\sqrt{S^*}}}(\boldsymbol{x}_{a_2}, \boldsymbol{x}_{b_2}), \ldots, \mathcal{D}_{\boldsymbol{\varphi}, \mathcal{V}_{S^*}^{\sqrt{S^*}}}(\boldsymbol{x}_{a_{K-1}}, \boldsymbol{x}_{b_{K-1}})\}.
\end{aligned}
\tag{0.23}
$$

By further constructing Eq. (0.20) $K-1$ times for $(a_{K-1}, b_{K-1}), (a_{K-2}, b_{K-2}), \ldots, (a_1, b_1)$, we finally have that

$$
\mathcal{D}_{\boldsymbol{\varphi}, \mathcal{V}^*}(\boldsymbol{x}_{a_{K+1}}, \boldsymbol{x}_{b_{K+1}}) \leq \mathcal{D}_{\boldsymbol{\varphi}, \mathcal{V}^*}(\boldsymbol{x}_{a_K}, \boldsymbol{x}_{b_K}) \leq \ldots \leq \mathcal{D}_{\boldsymbol{\varphi}, \mathcal{V}^*}(\boldsymbol{x}_{a_1}, \boldsymbol{x}_{b_1}),
\tag{0.24}
$$

which implies that the inequality holds for $(a_1, b_1) \succcurlyeq (a_2, b_2) \succcurlyeq \ldots \succcurlyeq (a_{K+1}, b_{K+1})$.

**iii).** By integrating i) and ii), we have that $\mathcal{D}_{\boldsymbol{\varphi}, \mathcal{V}^*}(\boldsymbol{x}_{a_1}, \boldsymbol{x}_{b_1}) \geq \mathcal{D}_{\boldsymbol{\varphi}, \mathcal{V}^*}(\boldsymbol{x}_{a_2}, \boldsymbol{x}_{b_2}) \geq \cdots \geq \mathcal{D}_{\boldsymbol{\varphi}, \mathcal{V}^*}(\boldsymbol{x}_{\mathrm{C}_N^2}, \boldsymbol{x}_{\mathrm{C}_N^2})$ can be satisfied for $(a_1, b_1) \succcurlyeq (a_2, b_2) \succcurlyeq \cdots \succcurlyeq (a_{\mathrm{C}_N^2}, b_{\mathrm{C}_N^2})$. The proof is completed. $\qquad\square$

## B.4 Proof for Theorem 4

*Proof.* We firstly show that for the given dataset $\mathscr{X}$, there exist positive constants $V_{\min}$ and $D_{\min}$, such that the learned encoder $\boldsymbol{\varphi}^*$ and measure-head $\mathcal{H}^*$ (corresponding to the volume function $\mathcal{V}^*$)

$$
\mathcal{V}^*(\boldsymbol{x}_i) \geq V_{\min}, \ \forall i = 1, 2, \ldots, N,
\tag{0.25}
$$

and

$$
\mathcal{D}_{\boldsymbol{\varphi}^*, \mathcal{V}^*}(\boldsymbol{x}_i, \boldsymbol{x}_j) \geq D_{\min}, \ 1 \leq i < j \leq N.
\tag{0.26}
$$

This is because that $\mathcal{F}(\boldsymbol{\varphi}^*, \mathcal{H}^*) \leq \mathcal{F}(\overline{\boldsymbol{\varphi}}, \overline{\mathcal{H}})$, namely

$$
\begin{aligned}
&\mathbb{E}_{\boldsymbol{x}, \{b_j\}_{j=1}^n}\left[-\log\frac{e^{-\mathcal{D}_{\boldsymbol{\varphi}^*, \mathcal{H}^*}(\boldsymbol{x}, \boldsymbol{x}^+)/\gamma}}{e^{-\mathcal{D}_{\boldsymbol{\varphi}^*, \mathcal{H}^*}(\boldsymbol{x}, \boldsymbol{x}^+)/\gamma} + \sum_{j=1}^n e^{-\mathcal{D}_{\boldsymbol{\varphi}^*, \mathcal{H}^*}(\boldsymbol{x}, \boldsymbol{x}_{b_j})/\gamma}}\right] + \lambda\mathbb{E}_{\{b_j\}_{j=1}^n}\left[\sum_{j=1}^n e^{-\mathcal{V}^*(\boldsymbol{x}_{b_j})}\right] \\
&\leq \mathbb{E}_{\boldsymbol{x}, \{b_j\}_{j=1}^n}\left[-\log\frac{e^{-\mathcal{D}_{\overline{\boldsymbol{\varphi}}, \overline{\mathcal{H}}}(\boldsymbol{x}, \boldsymbol{x}^+)/\gamma}}{e^{-\mathcal{D}_{\overline{\boldsymbol{\varphi}}, \overline{\mathcal{H}}}(\boldsymbol{x}, \boldsymbol{x}^+)/\gamma} + \sum_{j=1}^n e^{-\mathcal{D}_{\overline{\boldsymbol{\varphi}}, \overline{\mathcal{H}}}(\boldsymbol{x}, \boldsymbol{x}_{b_j})/\gamma}}\right] + \lambda\mathbb{E}_{\{b_j\}_{j=1}^n}\left[\sum_{j=1}^n e^{-\overline{\mathcal{V}}(\boldsymbol{x}_{b_j})}\right],
\end{aligned}
\tag{0.27}
$$

and thus

$$
\begin{aligned}
&e^{-\mathcal{V}^*(\boldsymbol{x}_{b_k})} \\
&\leq \mathbb{E}_{\boldsymbol{x}, \{b_j\}_{j=1}^n}\left[-\log\frac{e^{-\mathcal{D}_{\overline{\boldsymbol{\varphi}}, \overline{\mathcal{H}}}(\boldsymbol{x}, \boldsymbol{x}^+)/\gamma}}{e^{-\mathcal{D}_{\overline{\boldsymbol{\varphi}}, \overline{\mathcal{H}}}(\boldsymbol{x}, \boldsymbol{x}^+)/\gamma} + \sum_{j=1}^n e^{-\mathcal{D}_{\overline{\boldsymbol{\varphi}}, \overline{\mathcal{H}}}(\boldsymbol{x}, \boldsymbol{x}_{b_j})/\gamma}} + \log\frac{e^{-\mathcal{D}_{\boldsymbol{\varphi}^*, \mathcal{H}^*}(\boldsymbol{x}, \boldsymbol{x}^+)/\gamma}}{e^{-\mathcal{D}_{\boldsymbol{\varphi}^*, \mathcal{H}^*}(\boldsymbol{x}, \boldsymbol{x}^+)/\gamma} + \sum_{j=1}^n e^{-\mathcal{D}_{\boldsymbol{\varphi}^*, \mathcal{H}^*}(\boldsymbol{x}, \boldsymbol{x}_{b_j})/\gamma}}\right] \\
&\quad + \lambda\mathbb{E}_{\{b_j\}_{j=1}^n}\left[\sum_{j=1}^n e^{-\overline{\mathcal{V}}(\boldsymbol{x}_{b_j})}\right] - \lambda\mathbb{E}_{\{b_j\}_{j=1, j \neq k}^n}\left[\sum_{j=1}^n e^{-\mathcal{V}^*(\boldsymbol{x}_{b_j})}\right],
\end{aligned}
\tag{0.28}
$$

so that

$$
\begin{aligned}
&\mathcal{V}^*(\boldsymbol{x}_{b_k}) \\
&\geq -\log\left[\mathbb{E}_{\boldsymbol{x}, \{b_j\}_{j=1}^n}\left[-\log\frac{e^{-\mathcal{D}_{\overline{\boldsymbol{\varphi}}, \overline{\mathcal{H}}}(\boldsymbol{x}, \boldsymbol{x}^+)/\gamma}}{e^{-\mathcal{D}_{\overline{\boldsymbol{\varphi}}, \overline{\mathcal{H}}}(\boldsymbol{x}, \boldsymbol{x}^+)/\gamma} + \sum_{j=1}^n e^{-\mathcal{D}_{\overline{\boldsymbol{\varphi}}, \overline{\mathcal{H}}}(\boldsymbol{x}, \boldsymbol{x}_{b_j})/\gamma}} + \log\frac{e^{-\mathcal{D}_{\boldsymbol{\varphi}^*, \mathcal{H}^*}(\boldsymbol{x}, \boldsymbol{x}^+)/\gamma}}{e^{-\mathcal{D}_{\boldsymbol{\varphi}^*, \mathcal{H}^*}(\boldsymbol{x}, \boldsymbol{x}^+)/\gamma} + \sum_{j=1}^n e^{-\mathcal{D}_{\boldsymbol{\varphi}^*, \mathcal{H}^*}(\boldsymbol{x}, \boldsymbol{x}_{b_j})/\gamma}}\right]\right. \\
&\quad \left. + \lambda\mathbb{E}_{\{b_j\}_{j=1}^n}\left[\sum_{j=1}^n e^{-\overline{\mathcal{V}}(\boldsymbol{x}_{b_j})}\right] - \lambda\mathbb{E}_{\{b_j\}_{j=1, j \neq k}^n}\left[\sum_{j=1}^n e^{-\mathcal{V}^*(\boldsymbol{x}_{b_j})}\right]\right] \\
&\geq \log\left[\mathbb{E}_{\boldsymbol{x}, \{b_j\}_{j=1}^n}\left[\log\frac{e^{-\mathcal{D}_{\boldsymbol{\varphi}^*, \mathcal{H}^*}(\boldsymbol{x}, \boldsymbol{x}^+)/\gamma}}{e^{-\mathcal{D}_{\boldsymbol{\varphi}^*, \mathcal{H}^*}(\boldsymbol{x}, \boldsymbol{x}^+)/\gamma} + \sum_{j=1}^n e^{-\mathcal{D}_{\boldsymbol{\varphi}^*, \mathcal{H}^*}(\boldsymbol{x}, \boldsymbol{x}_{b_j})/\gamma}} \frac{e^{-\mathcal{D}_{\overline{\boldsymbol{\varphi}}, \overline{\mathcal{H}}}(\boldsymbol{x}, \boldsymbol{x}^+)/\gamma} + \sum_{j=1}^n e^{-\mathcal{D}_{\overline{\boldsymbol{\varphi}}, \overline{\mathcal{H}}}(\boldsymbol{x}, \boldsymbol{x}_{b_j})/\gamma}}{e^{-\mathcal{D}_{\overline{\boldsymbol{\varphi}}, \overline{\mathcal{H}}}(\boldsymbol{x}, \boldsymbol{x}^+)/\gamma}}\right]\right. \\
&\quad \left. + \frac{1}{\lambda\mathbb{E}_{\{b_j\}_{j=1}^n}\left[\sum_{j=1}^n e^{-\overline{\mathcal{V}}(\boldsymbol{x}_{b_j})}\right] - \lambda\mathbb{E}_{\{b_j\}_{j=1, j \neq k}^n}\left[\sum_{j=1}^n e^{-\mathcal{V}^*(\boldsymbol{x}_{b_j})}\right]}\right].
\end{aligned}
\tag{0.29}
$$

So we easily have that

$$\mathcal{V}^*(\boldsymbol{x}_{b_k})$$

$$\geq \log\left[\frac{1}{\lambda\mathbb{E}_{\{b_j\}_{j=1}^n}\left[\sum_{j=1}^n \mathrm{e}^{-\overline{\mathcal{V}}(\boldsymbol{x}_{b_j})}\right] - \lambda\mathbb{E}_{\{b_j\}_{j=1,j\neq k}^n}\left[\sum_{j=1}^n \mathrm{e}^{-\mathcal{V}^*(\boldsymbol{x}_{b_j})}\right]}\right]$$

$$\geq \log\left[\frac{1}{\lambda\mathrm{e}^{-\overline{\mathcal{V}}(\boldsymbol{x}_{b_k})}}\right]$$

$$= \log\left[\frac{1}{\lambda}\mathrm{e}^{\overline{\mathcal{V}}(\boldsymbol{x}_{b_k})}\right]$$

$$= \log[\frac{1}{\lambda}] + \log\left[\mathrm{e}^{\overline{\mathcal{V}}(\boldsymbol{x}_{b_k})}\right]$$

$$= \frac{1}{\lambda} + \overline{\mathcal{V}}(\boldsymbol{x}_{b_k}) = V_{\min}. \tag{0.30}$$

Meanwhile, we have

$$\mathbb{E}_{\boldsymbol{x},\{b_j\}_{j=1}^n}\left[\log\frac{\mathrm{e}^{-\mathcal{D}_{\boldsymbol{\varphi}^*,\mathcal{H}^*}(\boldsymbol{x},\boldsymbol{x}^+)/\gamma}+\sum_{j=1}^n \mathrm{e}^{-\mathcal{D}_{\boldsymbol{\varphi}^*,\mathcal{H}^*}(\boldsymbol{x},\boldsymbol{x}_{b_j})/\gamma}}{\mathrm{e}^{-\mathcal{D}_{\boldsymbol{\varphi}^*,\mathcal{H}^*}(\boldsymbol{x},\boldsymbol{x}^+)/\gamma}}\right]$$

$$\leq \mathbb{E}_{\boldsymbol{x},\{b_j\}_{j=1}^n}\left[-\log\frac{\mathrm{e}^{-\mathcal{D}_{\overline{\boldsymbol{\varphi}},\overline{\mathcal{H}}}(\boldsymbol{x},\boldsymbol{x}^+)/\gamma}}{\mathrm{e}^{-\mathcal{D}_{\overline{\boldsymbol{\varphi}},\overline{\mathcal{H}}}(\boldsymbol{x},\boldsymbol{x}^+)/\gamma}+\sum_{j=1}^n \mathrm{e}^{-\mathcal{D}_{\overline{\boldsymbol{\varphi}},\overline{\mathcal{H}}}(\boldsymbol{x},\boldsymbol{x}_{b_j})/\gamma}}\right] + \lambda\mathbb{E}_{\{b_j\}_{j=1}^n}\left[\sum_{j=1}^n \mathrm{e}^{-\overline{\mathcal{V}}(\boldsymbol{x}_{b_j})}\right]$$

$$- \lambda\mathbb{E}_{\{b_j\}_{j=1}^n}\left[\sum_{j=1}^n \mathrm{e}^{-\mathcal{V}^*(\boldsymbol{x}_{b_j})}\right]$$

$$= E_1, \tag{0.31}$$

which implies that

$$\mathrm{e}^{-\mathcal{D}_{\boldsymbol{\varphi}^*,\mathcal{H}^*}(\boldsymbol{x},\boldsymbol{x}_{b_j})/\gamma}$$

$$\leq \mathrm{e}^{-\mathcal{D}_{\boldsymbol{\varphi}^*,\mathcal{H}^*}(\boldsymbol{x},\boldsymbol{x}^+)/\gamma}\exp(E_1) - (\mathrm{e}^{-\mathcal{D}_{\boldsymbol{\varphi}^*,\mathcal{H}^*}(\boldsymbol{x},\boldsymbol{x}^+)/\gamma}+\sum_{j\neq k}^n \mathrm{e}^{-\mathcal{D}_{\boldsymbol{\varphi}^*,\mathcal{H}^*}(\boldsymbol{x},\boldsymbol{x}_{b_j})/\gamma}), \tag{0.32}$$

and thus

$$\mathcal{D}_{\boldsymbol{\varphi}^*,\mathcal{H}^*}(\boldsymbol{x},\boldsymbol{x}_{b_j})$$

$$\geq -\gamma\log\left[\mathrm{e}^{-\mathcal{D}_{\boldsymbol{\varphi}^*,\mathcal{H}^*}(\boldsymbol{x},\boldsymbol{x}^+)/\gamma}\exp(E_1) - (\mathrm{e}^{-\mathcal{D}_{\boldsymbol{\varphi}^*,\mathcal{H}^*}(\boldsymbol{x},\boldsymbol{x}^+)/\gamma}+\sum_{j\neq k}^n \mathrm{e}^{-\mathcal{D}_{\boldsymbol{\varphi}^*,\mathcal{H}^*}(\boldsymbol{x},\boldsymbol{x}_{b_j})/\gamma})\right]$$

$$= \gamma\log\left[\frac{1}{\mathrm{e}^{-\mathcal{D}_{\boldsymbol{\varphi}^*,\mathcal{H}^*}(\boldsymbol{x},\boldsymbol{x}^+)/\gamma}\exp(E_1) - (\mathrm{e}^{-\mathcal{D}_{\boldsymbol{\varphi}^*,\mathcal{H}^*}(\boldsymbol{x},\boldsymbol{x}^+)/\gamma}+\sum_{j\neq k}^n \mathrm{e}^{-\mathcal{D}_{\boldsymbol{\varphi}^*,\mathcal{H}^*}(\boldsymbol{x},\boldsymbol{x}_{b_j})/\gamma})}\right]$$

$$\geq \gamma\log\left[\frac{1}{\mathrm{e}^{-\mathcal{D}_{\boldsymbol{\varphi}^*,\mathcal{H}^*}(\boldsymbol{x},\boldsymbol{x}^+)/\gamma}}\right]$$

$$\geq \gamma\log\left[\frac{1}{\mathrm{e}^{-\mathcal{D}_{\boldsymbol{\varphi}^*,\mathcal{H}^*}(\boldsymbol{x},\boldsymbol{x}^+)/\gamma}}\right]$$

$$= \mathcal{D}_{\boldsymbol{\varphi}^*,\mathcal{H}^*}(\boldsymbol{x},\boldsymbol{x}^+) = D_{\min}. \tag{0.33}$$

Then we go ahead to show the finite coverage. To be specific, the union volume of all $N$ data balls $\mathcal{B}(\boldsymbol{x}_1, \mathcal{V}^*(\boldsymbol{x}_1)), \mathcal{B}(\boldsymbol{x}_2, \mathcal{V}^*(\boldsymbol{x}_2)), \ldots, \mathcal{B}(\boldsymbol{x}_N, \mathcal{V}^*(\boldsymbol{x}_N))$ can be calculated as

$$
\int_{\boldsymbol{z}} \operatorname{sign}[\boldsymbol{z} \in \bigcup_{i=1}^{N} \mathcal{B}(\boldsymbol{x}_i, \mathcal{V}(\boldsymbol{x}_i))] \mathrm{d}\boldsymbol{z}
$$

$$
= \sum_{i=1}^{N} \int_{\boldsymbol{z}} \operatorname{sign}[\boldsymbol{z} \in \mathcal{B}(\boldsymbol{x}_i, \mathcal{V}(\boldsymbol{x}_i))] \mathrm{d}\boldsymbol{z} - \sum_{i<j} \int_{\boldsymbol{z}} \operatorname{sign}[\boldsymbol{z} \in \mathcal{B}(\boldsymbol{x}_i, \mathcal{V}(\boldsymbol{x}_i)) \bigcap \boldsymbol{z} \in \mathcal{B}(\boldsymbol{x}_j, \mathcal{V}(\boldsymbol{x}_j))] \mathrm{d}\boldsymbol{z}
$$

$$
+ \sum_{i<j<k} \int_{\boldsymbol{z}} \operatorname{sign}[\boldsymbol{z} \in \mathcal{B}(\boldsymbol{x}_i, \mathcal{V}(\boldsymbol{x}_i)) \bigcap \boldsymbol{z} \in \mathcal{B}(\boldsymbol{x}_j, \mathcal{V}(\boldsymbol{x}_j)) \bigcap \boldsymbol{z} \in \mathcal{B}(\boldsymbol{x}_k, \mathcal{V}(\boldsymbol{x}_k))] \mathrm{d}\boldsymbol{z} - \ldots + \int_{\boldsymbol{z}} \operatorname{sign}[\boldsymbol{z} \in \bigcap_{i=1}^{N} \mathcal{B}(\boldsymbol{x}_i, \mathcal{V}(\boldsymbol{x}_i))] \mathrm{d}\boldsymbol{z}
$$

$$
\geq \sum_{i=1}^{N} \int_{\boldsymbol{z}} \operatorname{sign}[\boldsymbol{z} \in \mathcal{B}(\boldsymbol{x}_i, \mathcal{V}(\boldsymbol{x}_i))] \mathrm{d}\boldsymbol{z} - N \max_{i<j} \left( \int_{\boldsymbol{z}} \operatorname{sign}[\boldsymbol{z} \in \mathcal{B}(\boldsymbol{x}_i, \mathcal{V}(\boldsymbol{x}_i)) \bigcap \boldsymbol{z} \in \mathcal{B}(\boldsymbol{x}_j, \mathcal{V}(\boldsymbol{x}_j))] \mathrm{d}\boldsymbol{z} \right)
$$

$$
= N V_{\min} - N \mu(D_{min}) V_{\min}
$$

$$
= N(1 - \mu(D_{min})) V_{\min}, \tag{0.34}
$$

where $\mu(D_{min}) \in (0, 1)$. Finally, we let

$$
N = \left\lceil \frac{\rho |L - U|^m}{(1 - \mu(D_{min})) V_{\min}} \right\rceil, \tag{0.35}
$$

and have that

$$
\frac{\int_{\boldsymbol{z}} \operatorname{sign}[\boldsymbol{z} \in \bigcup_{i=1}^{N} \mathcal{B}(\boldsymbol{x}_i, \mathcal{V}(\boldsymbol{x}_i))] \mathrm{d}\boldsymbol{z}}{\int_{\boldsymbol{z} \in [L, U]^m} 1 \mathrm{d}\boldsymbol{z}}
$$

$$
\geq N(1 - \mu(D_{min})) V_{\min} \cdot \frac{1}{|L - U|^m}
$$

$$
\geq \frac{\rho |L - U|^m}{(1 - \mu(D_{min})) V_{\min}} (1 - \mu) V_{\min} \cdot \frac{1}{|L - U|^m}
$$

$$
= \rho, \tag{0.36}
$$

which completes the proof. $\qquad \square$

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
