# OpenReview forum: "Volume-Aware Distance for Robust Similarity Learning"
_ICML.cc/2025/Conference — ICML 2025 poster_

### Official Review · Reviewer_25dP · 2025-03-01

**Overall Recommendation:** 4

**Summary:**

This paper presents Volume-Aware Distance (VAD), a novel metric for similarity learning that extends traditional point-wise distances to field-to-field distances by introducing volume-aware data representations. The authors propose a measure-head network for volume prediction and a volume expansion regularizer to improve generalization. The method is mathematically well-grounded, providing strong theoretical guarantees, and is empirically validated across multiple domains.

**Claims And Evidence:**

The main contributions of the paper are: A new similarity metric (VAD) that enhances generalization by incorporating volume-awareness. A theoretical framework demonstrating VAD’s tighter generalization bounds. Empirical results showing VAD’s superiority over state-of-the-art similarity learning methods.

**Essential References Not Discussed:**

No critical references appear to be missing.

**Experimental Designs Or Analyses:**

The experiments are well-structured, covering both metric learning and contrastive learning. The baselines used for comparison are strong, ensuring credibility. Ablation studies demonstrate the necessity of each component.

**Methods And Evaluation Criteria:**

The proposed VAD framework, including the measure-head network and regularizer, is well-motivated. The evaluation includes multiple supervised and unsupervised tasks, making the results highly convincing.

**Other Comments Or Suggestions:**

None

**Other Strengths And Weaknesses:**

Strengths:

Novel and impactful contribution to similarity learning.

Solid theoretical underpinnings.

Comprehensive empirical evaluation.

Weakness:

The computational complexity of the method could be further analyzed in large-scale settings.

**Questions For Authors:**

See the weakness.

**Relation To Broader Scientific Literature:**

The work is well-grounded in existing literature and makes a meaningful contribution to the field of similarity learning.

**Theoretical Claims:**

The authors present rigorous mathematical proofs for generalization error bounds and sample-space coverage, which validate the effectiveness of VAD.

---

> ### Author Rebuttal · Authors · 2025-04-01
>
> Thank you for your positive and constructive comments! Our responses are given below.
>
> ---
>
> **Comment_1:** The computational complexity of the method could be further analyzed in large-scale settings.
>
> **Response_1:** Thank you for your suggestion! Here we would like to provide the training time comparison of our VADSL and the baseline methods (SimCLR and SwAV) on the large-scale ImageNet-1K dataset. Specifically, we use eight NVIDIA TeslaV100 GPUs to train our models based on SimCLR and SwAV with 100 epochs, respectively. For each case, we set the batch size to 512, 1024, and 1536.
>
> | Method | &nbsp; ImageNet-1K (ba. si. = 512) &nbsp; | &nbsp; ImageNet-1K (ba. si. = 1024) &nbsp; | &nbsp; ImageNet-1K (ba. si. = 1536) &nbsp; |
> | :--- | :----: | :----: | :----: |
> | SimCLR | 70.1 | 35.2 | 23.1 |
> | SwAV | 71.2 | 36.7 | 24.2 |
> | VADSL (SimCLR + VAD) | 71.5 | 35.6 | 23.5 |
> | VADSL (SwAV + VAD) | 72.1 | 36.9 | 24.6|
>
>
> The table above reveals that the introduction of the proposed VAD component causes very little increase in time consumption (less than **2%**). This is because the gradient calculation of VAD is independent of the size of the training data, thereby keeping the training time well within practical limits. We will also add the above time comparison in the camera-ready version if this paper is finally accepted.

---

### Official Review · Reviewer_EAHq · 2025-03-02

**Overall Recommendation:** 4

**Summary:**

The paper introduces Volume-Aware Distance (VAD), a novel metric for robust similarity learning. Unlike conventional point-level similarity measures, VAD models instances as volume-aware data balls, improving generalization by capturing field-to-field geometric relationships. The paper also proposes a measure-head network to learn instance volumes and a volume expansion regularizer to further enhance generalization. Theoretical analysis shows that VAD tightens generalization bounds and preserves topological properties, while extensive experiments on supervised and unsupervised tasks demonstrate clear advantages over state-of-the-art methods.

**Claims And Evidence:**

The key claims in the paper include: VAD generalizes better than conventional similarity learning methods, supported by theoretical proofs. VAD outperforms baseline models across multiple tasks, shown via strong empirical results.

**Essential References Not Discussed:**

No.

**Experimental Designs Or Analyses:**

Experiments cover multiple learning paradigms (metric learning and contrastive learning). Baseline comparisons are thorough, ensuring fairness. Ablation studies validate the necessity of each proposed component.

**Methods And Evaluation Criteria:**

The measure-head network and volume expansion regularizer are well-designed and logically justified. The proposed method is evaluated comprehensively across various datasets and baselines, making the results reliable.

**Other Comments Or Suggestions:**

None

**Other Strengths And Weaknesses:**

Strengths:
Well-motivated and novel approach to similarity learning.
Strong theoretical backing with comprehensive analysis.
Extensive empirical validation across diverse datasets.

Weaknesses:
Computational overhead due to volume estimation, though the authors argue it remains manageable.

**Questions For Authors:**

i). How sensitive is VAD to the choice of the measure-head network’s architecture?

ii). Would VAD work in few-shot learning settings where data volume is extremely limited?

**Relation To Broader Scientific Literature:**

The paper clearly situates itself within the literature on metric learning, contrastive learning, and regularization methods. The citations are appropriate and comprehensive.

**Theoretical Claims:**

I examined the proofs for generalization bounds and distance flexibility. The claims are mathematically sound, and the derivations align with existing principles in similarity learning.

---

> ### Author Rebuttal · Authors · 2025-04-01
>
> Thank you for your positive and constructive comments! Our point-by-point responses are provided below.
>
> ---
>
> **Comment_1:** Computational overhead due to volume estimation, though the authors argue it remains manageable.
>
> **Response_1:** Thanks for your comments! We agree with the reviewer that the volume estimation does indeed introduce additional computation burden due to the necessary feedforward of the head network. However, we want to clarify that such additional computation cost is actually necessary to achieve better performance and is negligible in real-world use.
> Specifically, as shown in Appendix A.3 of our manuscript, the baseline method SimCLR and our method VADSL spend 2.3 hours and 2.4 hours, respectively, to train 100 epochs on CIFAR-10 (batch size = 512). It implies that the calculation of volume estimation only adds less than **5%** of the time consumption. Similarly, on ImageNet-100, the time cost of the baseline method and our method are 10.9 hours and 11.2 hours, respectively, which means that the volume estimation merely requires an additional **3%** time consumption. In summary, we believe that the additional time cost of our method is relatively small and acceptable in practice.
>
> ---
>
> **Comment_2:** How sensitive is VAD to the choice of the measure-head network’s architecture?
>
> **Response_2:** Thanks for your comments! Since the measure-head network is implemented by the classical MLP with a single hidden layer, the only human-tuned parameter is the dimensionality of the hidden layer (because the input-feature dimensionality is usually fixed and the output-layer dimensionality is always 1). Therefore, here we want to investigate the robustness of our method by changing the hidden-layer dimensionality.
> |Dataset|&nbsp; 32-dim &nbsp;|&nbsp; 64-dim &nbsp;|&nbsp; 128-dim &nbsp;|&nbsp; 256-dim &nbsp;|&nbsp; 512-dim &nbsp;|
> | :--- | :----: | :----: | :----: | :----: | :----: |
> | CIFAR-10 (400-eps, 512-bs) | 86.3 | 93.5 | 94.9 | 93.5 | 91.2 |
> | STL-10 (400-eps, 512-bs) | 79.4 | 85.2 | 85.6 | 85.8 | 81.1 |
>
> In above table, we vary the hidden-layer dimensionality from 32 to 512 to record the final performance of our method on STL-10 and CIFAR-10. We can easily observe that the hidden-layer dimensionality indeed affects the classification accuracy to some extent, because VAD needs a certain amount of hidden nodes to ensure its nonlinearity. However, it is also obvious that the performance of our method is relatively stable around 128 (e.g., from 64 to 256), which means that such a hyper-parameter can be easily tuned in practical use.
> Meanwhile, we would like to further investigate the sensitivity of VAD by setting different number of hidden layers (including 1-layer, 2-layers, and 3-layer). For each case, we let the dimensionalities of all hidden layers be the same (i.e., 128-dim). The following table records the corresponding accuracy rates of our method on CIFAR-10 and STL-10.
> |Dataset|&nbsp; 1-layer hidden &nbsp;|&nbsp; 2-layers hidden &nbsp;|&nbsp; 3-layers hidden &nbsp;|
> | :--- | :----: | :----: | :----: |
> | CIFAR-10 (400-eps, 512-bs) | 94.9 | 93.6 | 94.5 |
> | STL-10 (400-eps, 512-bs) | 85.6 | 85.6 | 85.5 |
>
> We can clearly observe that the results on both datasets are relatively stable when the layer number changes, and this means that our method is robust to the choice of network architectures.
>
> ---
>
> **Comment_3:** Would VAD work in few-shot learning settings where data volume is extremely limited?
>
> **Response_3:** Thanks for your comments! Actually, VAD is able to work well for few-shot recognition tasks. To be specific, for the four datasets (i.e., CAR-196, CUB-200, SOP, and In-Shop) used in the metric learning task, no classes of the test data are involved in the training phase, and only very few reference examples are provided for classification during the test phase. In such a few-shot setting, our VAD effectively improves the NMI and Recall@K scores of the baseline methods, and our final performance can surpass SOTA methods in most cases (Tab. 2). We believe that the effectiveness of our VAD in few-shot learning settings is well validated.

---

### Official Review · Reviewer_cnEk · 2025-03-03

**Overall Recommendation:** 3

**Summary:**

This paper introduces a novel approach to similarity learning with the Volume-Aware Distance (VAD) metric. Instead of relying on traditional point-level similarity measures, VAD models data instances as volume-aware data spheres, allowing it to capture both observed and unobserved neighbor relationships. To improve generalization, the method utilizes a measure-head network for estimating instance volumes and incorporates a volume expansion regularizer. Theoretical analysis confirms its enhanced error bounds and topological properties, while extensive experiments across multiple domains demonstrate its effectiveness compared to existing methods.

**Claims And Evidence:**

This paper asserts that VAD enhances generalization by encompassing a wider sample space while maintaining essential topological properties, as substantiated by theoretical analysis. Furthermore, VAD surpasses leading methods in both supervised metric learning and unsupervised contrastive learning. These conclusions are reinforced by rigorous mathematical derivations and comprehensive experimental evaluations, highlighting its robustness and broad applicability.

**Essential References Not Discussed:**

Not obvious

**Experimental Designs Or Analyses:**

The experimental design covers both supervised and unsupervised settings. The baselines are well-selected, and results consistently show the advantage of the proposed VAD. The t-SNE visualizations and ablation studies further strengthen the empirical support.

**Methods And Evaluation Criteria:**

The experimental setup is thorough, covering multiple datasets (e.g., CIFAR-10, ImageNet, SOP, CUB-200) and different learning paradigms. The proposed method makes sense in similarity learning, particularly in contrastive learning and metric learning.

**Other Comments Or Suggestions:**

N/A

**Other Strengths And Weaknesses:**

Strengths:
+ Novel extension of distance metric learning and comprehensive theoretical guarantees.
+ Comprehensive empirical evaluation. Well-written and clearly structured.

Weaknesses:

Some details are unclear:

a. Can VAD be extended beyond similarity learning, e.g., in clustering?

b. How does the measure-head network perform when used with transformers instead of CNNs?

**Questions For Authors:**

a. Can VAD be extended beyond similarity learning, e.g., in clustering?

b. How does the measure-head network perform when used with transformers instead of CNNs?

**Relation To Broader Scientific Literature:**

The paper effectively connects its contributions to prior work in metric learning, contrastive learning, and regularization techniques. The citations are comprehensive and demonstrate a good understanding of the field.

**Theoretical Claims:**

The proofs related to generalization error bounds and distance flexibility looks fine. They are mathematically sound, and the theoretical contribution is clear.

---

> ### Author Rebuttal · Authors · 2025-04-01
>
> Thank you for your appreciation of the novelty, theoretical analyses, and experimental results of our paper! Thanks also for your very insightful and constructive suggestions! Our point-by-point responses are as follows.
>
> ---
>
> **Comment_1:** Can VAD be extended beyond similarity learning, e.g., in clustering?
>
> **Response_1:** Thanks for your comments! Yes, VAD is also applicable to clustering task. Actually, in our manuscript, we have already evaluated the clustering performance of VAD by showing the NMI scores on CAR-196, CUB-200, SOP, and In-Shop, where our VAD can successfully improve the baseline methods Npair and ProxyAnchor with at least 3 percentages (see Tab. 2). Here we also want to follow the reviewer’s suggestion to further validate the effectiveness of our VAD on the deep clustering model “contrastive clustering (CC)” (Li et al., Contrastive Clustering, AAAI’21). We simply implement our measure-head network by replacing the contrastive head in CC, so that we can calculate the corresponding VAD similarity values. We record the clustering accuracy and NMI scores of compared approaches on CIFAR-10 and CIFAR-100, and the following table reveals that our method still improves the strong baseline upon CC itself.
> |Setting|&nbsp; CIFAR-10 (Acc) &nbsp;|&nbsp; CIFAR-10 (NMI) &nbsp;|&nbsp;  CIFAR-100 (Acc) &nbsp;|&nbsp; CIFAR-100 (NMI) &nbsp;|
> | :--- | :----: | :----: | :----: | :----: |
> | CC (baseline) | 79.0 | 70.5 | 42.9 | 43.1 |
> | VADSL (ours) | 79.4 | 71.2 | 43.6 | 43.8 |
>
> ---
>
> **Comment_2:** How does the measure-head network perform when used with transformers instead of CNNs?
>
> **Response_2:** Thanks for your comments! We agree with the reviewer that transformer has shown very promising reliability in lots of vision tasks. As our Tab. 4 has provided the detailed performance results (i.e., k-NN, Top-1, Top-5), we would like to discuss here again the superiority of our method when it is equipped with the ViT model (ViT.16/48MB).
> Specifically, for both ImageNet-100 and ImageNet-1K, we implement our method on the cluster-free method BYOL as well as the cluster-based method SwAV, and the encoder networks of all compared methods are ViT.16. It can be observed that our method consistently improves the two baseline methods on all cases of the two datasets.  Furthermore, we also compare our method with SOTA methods including DINO, iBOT, and MTE, and we can find that our VADSL outperforms the SOTA in most cases (see the bottom of Tab. 4). This implies that our method works well with the vision transformer in the feature extraction, and also shows that our VAD is a fairly general technique that can be used in different frameworks.

---

### Official Review · Reviewer_P61C · 2025-03-07

**Overall Recommendation:** 4

**Summary:**

The paper introduces Volume-Aware Distance (VAD), a new similarity metric that generalizes conventional point-wise distance by incorporating volume information. The authors propose a measure-head network to learn instance volumes and a volume expansion regularizer to improve generalization. Theoretical analyses prove a tighter generalization bound, and experimental results show superior performance in metric learning and contrastive learning tasks.

**Claims And Evidence:**

The authors claim:VAD provides better generalization (supported by theoretical proofs).VAD outperforms state-of-the-art methods in supervised and unsupervised tasks (validated via extensive experiments). These claims are well-supported by rigorous proofs and strong empirical evidence.

**Essential References Not Discussed:**

No major missing references.

**Experimental Designs Or Analyses:**

Experiments are comprehensive, covering both supervised and unsupervised learning. Ablation studies clearly show the importance of each component. Comparison with strong baselines ensures credibility.

**Methods And Evaluation Criteria:**

The method is well-motivated and logically sound. Evaluation is extensive, with results on various domains and datasets, ensuring robustness.

**Other Comments Or Suggestions:**

NA

**Other Strengths And Weaknesses:**

Strengths: 1) Well-motivated and novel approach. 2) Strong theoretical guarantees.
3) Impressive empirical results.

Weaknesses:
Slightly higher computational cost due to volume estimation.

**Questions For Authors:**

(1)	Can VAD be applied to NLP tasks such as sentence similarity?

(2)	How does VAD behave in extremely high-dimensional spaces?

**Relation To Broader Scientific Literature:**

The paper relates well to existing literature in metric learning and contrastive learning.

**Theoretical Claims:**

The proofs for error bounds, flexibility, and sample-space coverage are well-structured and mathematically valid.

---

> ### Author Rebuttal · Authors · 2025-04-01
>
> Thanks for your positive and insightful comments! Our explanation and clarification can be found as follows.
>
> ---
>
> **Comment_1:** Slightly higher computational cost due to volume estimation.
>
> **Response_1:** Thanks for your comments! We agree with the reviewer that the volume estimation does indeed introduce additional computation burden due to the necessary feedforward of the head network. However, we want to clarify that such additional computation cost is actually necessary to achieve better performance and is negligible in real-world use.
> Specifically, as shown in Appendix A.3 of our manuscript, the baseline method SimCLR and our method VADSL spend 2.3 hours and 2.4 hours, respectively, to train 100 epochs on CIFAR-10 (batch size = 512). It implies that the calculation of volume estimation only adds less than **5%** of the time consumption. Similarly, on ImageNet-100, the time cost of the baseline method and our method are 10.9 hours and 11.2 hours, respectively, which means that the volume estimation merely requires an additional **3%** time consumption. In summary, we believe that the additional time cost of our method is relatively small and acceptable in practice.
>
> ---
>
> **Comment_2:** Can VAD be applied to NLP tasks such as sentence similarity?
>
> **Response_2:** Thanks for your comments! Yes, VAD can be applied to NLP tasks. In fact, we have already validated the effectiveness of our method on two sentence embedding related datasets including STS and BookCorpus. As the reviewer considered, for each dataset, we construct the positive (similar) pairs by aligning each sentence with its contextual sentences in the same paragraph, while we build the negative (dissimilar) pairs by combining those sentences from different paragraphs. For both datasets, our method can consistently improve the baseline methods and outperform the SOTA methods in most cases (see Tab. 5 and Fig. A1), so we believe that our method is applicable to the sentence embedding tasks.
>
> ---
>
> **Comment_3:** How does VAD behave in extremely high-dimensional spaces?
>
> **Response_3:** Thanks for your comments! Here we follow the reviewer’s suggestion to investigate the reliability of VAD for high-dimensional features. Specifically, we increase the output-layer dimensionality of the feature encoder from 512 to 4096, and record the corresponding accuracy of our method on STL-10 (batch size = 512). From the following table, we can clearly observe that the high-dimensional space indeed weakens the feature discriminability, where the accuracy decreases when the feature dimensionality is higher than 1024. However, in these high-dimensional cases, our VAD can still successfully improve the final performance on STL-10, which demonstrates the usefulness of VAD even in the high-dimensional space.
> |Setting|&nbsp; STL_100-eps (w/o VAD) &nbsp;|&nbsp; STL_400-eps (w/o VAD) &nbsp; |&nbsp;  STL_100-eps (w/ VAD) &nbsp;|&nbsp;  STL_400-eps (w/ VAD) &nbsp;|
> | :--- | :----: | :----: | :----: | :----: |
> | 512-dimension | 71.1±1.2 | 78.2±3.3 | &nbsp; &nbsp; **77.1±2.9** ✔| &nbsp; &nbsp; **85.6±3.2** ✔ |
> | 1024-dimension | 70.4±4.2 | 78.2±2.1 | &nbsp; &nbsp; **77.2±3.6** ✔ | &nbsp; &nbsp; **85.9±1.6** ✔ |
> | 2048-dimension | 68.3±2.3 | 76.5±1.3 | &nbsp; &nbsp; **73.3±1.2** ✔ | &nbsp; &nbsp; **82.3±2.3** ✔ |
> | 4096-dimension | 68.9±4.1 | 77.2±1.6 | &nbsp; &nbsp; **75.4±3.2** ✔ | &nbsp; &nbsp; **83.2±2.9** ✔ |

---

### Decision · Program_Chairs · 2025-05-01

**Decision:**

Accept (poster)

**Comment:**

This paper introduces Volume-Aware Distance (VAD), a novel metric for robust similarity learning. Unlike conventional point-level similarity measures, VAD models instances as volume-aware data balls, improving generalization by capturing field-to-field geometric relationships.

This paper received universal positive scores after the rebuttal period. All reviewers agreed that the approach is novel and that the empirical results were very solid.